# *EPHX1* mutations cause a lipoatrophic diabetes syndrome due to impaired epoxide hydrolysis and increased cellular senescence

Jeremie Gautheron[1,2], Christophe Morisseau[3], Wendy K Chung[4,5], Jamila Zammouri[1,2], Martine Auclair[1,2], Genevieve Baujat[6], Emilie Capel[1,2], Celia Moulin[1,2], Yuxin Wang[3], Jun Yang[3], Bruce D Hammock[3], Barbara Cerame[7], Franck Phan[2,8,9], Bruno Fève[1,2,10], Corinne Vigouroux[1,2,10,11], Fabrizio Andreelli[2,8,9], Isabelle Jeru[1,2,11]*

[1]Sorbonne Université-Inserm UMRS_938, Centre de Recherche Saint-Antoine (CRSA), Paris, France; [2]Institute of Cardiometabolism and Nutrition (ICAN), CHU Pitié-Salpêtrière - Saint-Antoine, Assistance Publique-Hôpitaux de Paris (AP-HP), Paris, France; [3]Department of Entomology and Nematology, and UC Davis Comprehensive Cancer Center, University of California, Davis, Davis, United States; [4]Department of Pediatrics, Columbia University Irving Medical Center, New York, United States; [5]Deparment of Medicine, Columbia University Irving Medical Center, New York, United States; [6]Service de Génétique Clinique, Hôpital Necker-Enfants Malades, AP-HP, Paris, France; [7]Goryeb Children's Hospital, Atlantic Health Systems, Morristown Memorial Hospital, Morristown, United States; [8]Service de Diabétologie-Métabolisme, Hôpital Pitié-Salpêtrière, AP-HP, Paris, France; [9]Sorbonne Université-Inserm UMRS_1269, Paris, France; [10]Centre National de Référence des Pathologies Rares de l'Insulino-Sécrétion et de l'Insulino-Sensibilité (PRISIS), Service de Diabétologie et Endocrinologie de la Reproduction, Hôpital Saint-Antoine, AP-HP, Paris, France; [11]Laboratoire commun de Biologie et Génétique Moléculaires, Hôpital Saint-Antoine, AP-HP, Paris, France

*For correspondence:
isabelle.jeru@aphp.fr

Competing interests: The authors declare that no competing interests exist.

**Abstract** Epoxide hydrolases (EHs) regulate cellular homeostasis through hydrolysis of epoxides to less-reactive diols. The first discovered EH was EPHX1, also known as mEH. EH functions remain partly unknown, and no pathogenic variants have been reported in humans. We identified two de novo variants located in EPHX1 catalytic site in patients with a lipoatrophic diabetes characterized by loss of adipose tissue, insulin resistance, and multiple organ dysfunction. Functional analyses revealed that these variants led to the protein aggregation within the endoplasmic reticulum and to a loss of its hydrolysis activity. CRISPR-Cas9-mediated *EPHX1* knockout (KO) abolished adipocyte differentiation and decreased insulin response. This KO also promoted oxidative stress and cellular senescence, an observation confirmed in patient-derived fibroblasts. Metreleptin therapy had a beneficial effect in one patient. This translational study highlights the importance of epoxide regulation for adipocyte function and provides new insights into the physiological roles of EHs in humans.

## Introduction

Epoxide hydrolases (EHs) constitute a small protein family, first characterized as a group of detoxifying enzymes (*Oesch et al., 1973*). The first EHs were identified more than 40 years ago, and, to date, five genes encoding EH have been identified in humans (*Decker et al., 2012*; *Fretland and Omiecinski, 2000*). Human microsomal and soluble EHs (mEH and sEH), also named EPHX1 and EPHX2, respectively, are the best known EHs. Structurally, they are alpha/beta hydrolase fold enzymes. They catalyze the rapid hydrolysis of epoxides, which are three membered cyclic ethers, to less-reactive and readily excretable diols (*Oesch et al., 1971*). In mammals, epoxides arise from cytochrome P450 (CYP450) oxidative metabolism of both xenobiotics and endogenous compounds (*El-Sherbeni and El-Kadi, 2014*). Epoxides from xenobiotics are involved in the development of cancers and organ damage through interaction with DNA, lipids, and proteins (*Nebert and Dalton, 2006*), and EHs are necessary for their detoxification. On the other hand, epoxides derived from endogenous fatty acids, called epoxy fatty acids (EpFAs), are important regulatory lipid mediators. They were shown to mediate several biological processes, including inflammation, angiogenesis, vasodilation, and nociception. By regulating EpFAs levels, EHs play a key role in regulating crucial signaling pathways for cellular homeostasis (*McReynolds et al., 2020*), and altered levels of EpFAs are associated with many disorders (*Morisseau and Hammock, 2013*). Although a large amount of work has been made to characterize EH functions, especially through the use of specific inhibitors and animal models, the full spectrum of their substrates and associated biological functions in human remain partly unknown (*Gautheron and Jéru, 2020*). It is also a challenge to clearly define the contribution of each EH in human disorders.

Association studies suggested a role of several single-nucleotide polymorphisms (SNPs) identified in *EPHX1* [MIM132810] and *EPHX2* [MIM132811] in multiple conditions, including liver cirrhosis, alcohol dependence, Crohn's disease, chronic obstructive pulmonary disease, preeclampsia, diabetes mellitus, and many cancers (*El-Sherbeni and El-Kadi, 2014*; *Václavíková et al., 2015*). This underscores the pleiotropic and crucial role of EHs in cell homeostasis, but no high-effect variants have been reported to date. In the present study, we identified two unrelated patients with a complex lipoatrophic and neurodevelopmental syndrome with severe metabolic manifestations and carrying a de novo variant in *EPHX1* identified by whole-exome sequencing (WES). Lipoatrophic diabetes, also known as lipodystrophic syndromes, are characterized by clinical lipoatrophy due to a defect in adipose tissue storage of triglycerides. This results in ectopic lipid infiltration of non-adipose tissues leading to insulin resistance, increased liver glucose production, hypertriglyceridemia, and liver steatosis. About 30 genes have been implicated in lipoatrophic diabetes (*Brown et al., 2016*). Although these disorders remain genetically unexplained in the vast majority of cases, there is growing interest in identifying their molecular and cellular bases to improve genetic counseling and personalize treatment (*Letourneau and Greeley, 2018*; *Sollier et al., 2020*).

EPHX1 is widely expressed with highest expression in the liver, adipose tissue, and adrenal glands (*Coller et al., 2001*). It was the first mammalian EH to be identified and was first purified from rabbit liver in the 1970s. It is an evolutionarily highly conserved biotransformation enzyme retained in microsomal membranes of the endoplasmic reticulum (ER) (*Coller et al., 2001*). For a long time, EPHX1 and EPHX2 were thought to simply fulfill distinct complementary roles (*Decker et al., 2009*). On the one hand, EPHX1 was well recognized to detoxify xenobiotic epoxides. On the other hand, EPHX2 was shown to hydrolyze endogenous terpenoid epoxides and EpFAs (*El-Sherbeni and El-Kadi, 2014*). More recently, EPHX1 was also shown in animal models to play a significant role in the hydrolysis of different endogenous EpFAs derived from polyunsaturated fatty acids (*Edin et al., 2018*; *Snider et al., 2007*; *Marowsky et al., 2017*; *Blum et al., 2019*). These EpFAs, including epoxyeicosatrienoic acids (EETs) and epoxyoctadecenoic acids (also called EpOMEs), are formed by ER-attached CYP450s and hydrolyzed by EPHX1 to their respective diols, the so-called dihydroxyeicosatrienoic acids (DHETs) and dihydroxyoctadecenoic acids (DiHOMEs), respectively. Nevertheless, from an evolutionary perspective, EPHX1 and EPHX2 are very different enzymes, which can be differentially inhibited by distinct chemical inhibitors, and display only a partially overlapping substrate selectivity.

The *EPHX1* variants identified in the studied patients are localized in the catalytic domain of the enzyme and were predicted to be pathogenic. This prompted us to assess their functional

consequences in several cellular models. The impact of the loss of EPHX1 activity on adipocyte differentiation and function was evaluated by developing CRISPR-Cas9-mediated genome-editing approaches.

## Results

### Identification of *EPHX1* variants in two unrelated patients

To identify novel genetic causes responsible for lipoatrophic diabetes, WES was carried out on a parent–offspring trio. The index case, patient 1, is a 25-year-old woman originating from Mauritania. Her disease phenotype was not explained by variants in genes known to be involved in lipoatrophic diabetes, as assessed by the analysis of a panel of genes used in routine genetic diagnosis (*Jéru et al., 2019*). The analysis of exome data led to the identification of a heterozygous variant in exon 7 of *EPHX1* (*Retterer et al., 2016*), which was subsequently confirmed by Sanger sequencing: c.997A>C, p.(Thr333Pro) (NM_000120.4) (*Figure 1A,B*). This variant was found to be de novo after paternity confirmation by genotyping a set of polymorphic markers. Due to the key role of *EPHX1* in cellular homeostasis and fatty acid metabolism (*Edin et al., 2018*), this gene was a good candidate. We then looked for additional individuals carrying a molecular defect in *EPHX1*. A second patient carrying a different de novo EPHX1 missense variant located in exon 9: c.1288G>C, p.(Gly430Arg) was identified through GeneMatcher (*Sobreira et al., 2015*; *Figure 1A,B*). Patient 2 is a 17-year-old woman originating from Western Europe and living in the United States. She also had an insulin-resistant lipoatrophic syndrome. We did not identify any alternative molecular etiology compatible with the disease phenotype in either of the two patients. A detailed list of the other rare de novo, compound heterozygous, and homozygous variants, as well as the reasons for their exclusion is provided in *Supplementary file 1* and *2*. The presence of chromosomal abnormalities was also previously excluded in the two patients by karyotype and SNP chromosome microarrays. Several additional lines of evidence supported the causal role of the two variants in the disease phenotype. These variants were absent from databases reporting variants from the general population (gnomAD, ExAC, dbSNP, and Exome Variant Server), as well as from ClinVar, a database that aggregates information about genomic variations and their relationship to human diseases. The variants identified herein affected amino acids strongly conserved throughout evolution, even in zebrafish and *Xenopus tropicalis* (*Figure 1—figure supplement 1*). The two variants predicted changes in the polarity of the corresponding amino acids, as well as in the charge for p.Gly430Arg. They were predicted pathogenic by all tested algorithms (PolyPhen-2, SIFT, CADD).

### Clinical features in patient 1

This patient (woman) was born at term after a normal pregnancy without intra-uterine growth retardation. The anthropometric parameters at birth were normal with a height of 49 cm and a weight of 2.8 kg. She was first referred for dysmorphic features including microcephaly with an occipito-frontal circumference (OFC) of 33 cm at birth (−1.5 SD), which remained present in adulthood with an OFC of 51 cm (−2.5 SD) at the age of 18 years. She also presented with a triangular-shaped face, prominent forehead, retrognathism, irregular and high hair line, high-arched palate, mandibulo-facial dysostosis including malar hypoplasia and retrognathism, teeth misalignments, arachnodactyly, camptodactyly, joint stiffness, and clitoromegaly (*Figure 1C,F* and *Table 1*). Patient 1 also displayed a progressive lipodystrophic phenotype with severe lipoatrophy of the face and limbs (*Figure 1C,G*). This lipoatrophic phenotype was further confirmed by dual X-ray absorptiometry (DXA) with a total fat mass of 15.8%, whereas the mean normal age-matched value is 31.4 ± 8.5% (*Imboden et al., 2017*), corresponding to a Z-score of −2.8. The study of segmental body composition revealed that the loss of adipose tissue was evenly distributed throughout the body (*Figure 1—figure supplement 2*). Her body mass index (BMI) was 20.2 kg/m$^2$ at the age of 25 years. The serum leptin levels, which are strongly correlated with total body fat mass, were very low in patient 1 (4 ng/mL) and similar to those usually reported in partial lipodystrophy (*Haque et al., 2002*), further confirming the lipoatrophic phenotype. Patient 1 was diagnosed with severe insulin-resistant diabetes at the age of 12 years, with hyperglycemia (44 mmol/L), and highly elevated HbA1c (18.9%). Insulin resistance was characterized by *acanthosis nigricans* in the neck, axilla, and back (*Figure 1E*), as well as by the very high insulin requirements (15 IU/kg/day) and by the low levels of total serum adiponectin (0.5 mg/L –

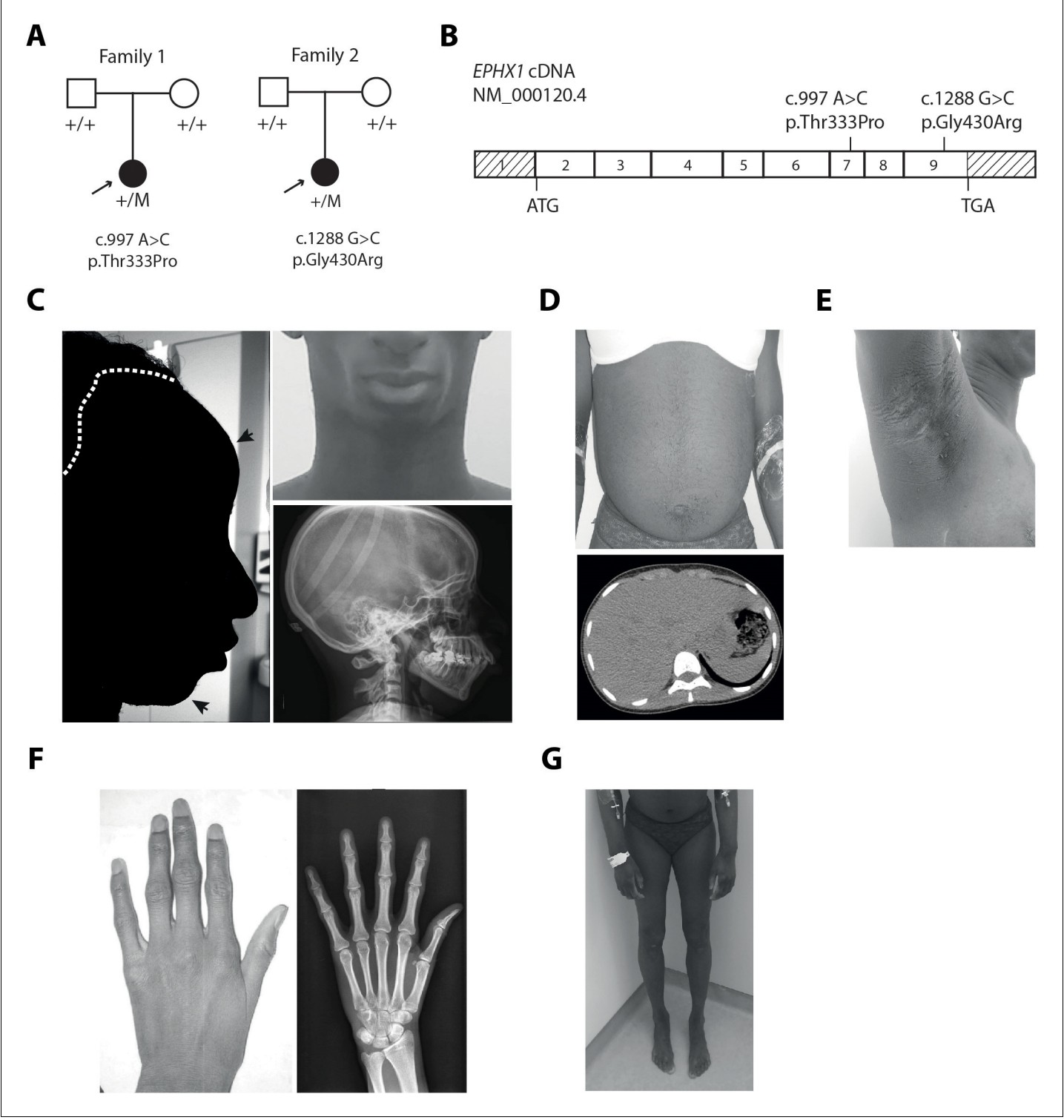

**Figure 1.** *EPHX1* pathogenic variants in a newly characterized lipoatrophic diabetes syndrome. (**A**) Genealogical trees and segregation analysis for the two *EPHX1* variants identified in this study. Arrows indicate probands. p.Thr333Pro and p.Gly430Arg were absent from both parents of each proband, indicating that they occurred de novo. +, normal allele; M, mutant allele. (**B**) Schematic of *EPHX1* transcript displaying the location of the two variants identified. (**C**) Characteristics of the patient's head. Left: Black shadow of the patient's profile over a grayscale photo. Black arrows point to frontal bossing and retrognathism. The white dotted line indicates the base of the scalp showing high hair line. Top right: Front photo of the patient's face showing lipoatrophy and retrognathism. Bottom right: Profile radiography of the skull showing teeth misalignments and mandibulo-facial dysostosis. (**D**) Top: Frontal photo of the patient's abdomen showing prominent abdomen with umbilical herniation and hirsutism. Bottom: axial computed

*Figure 1 continued on next page*

*Figure 1 continued*

tomography slice of the abdomen showing hepatomegaly and liver steatosis. (E) Picture of the armpit showing *acanthosis nigricans* and *molluscum pendulum*. (F) Picture (left) and radiography (right) of the left-hand showing arachnodactyly with tapered fingers and thickening of proximal interphalangeal joints. (G) Front picture of the legs showing distal lipoatrophy.

The online version of this article includes the following figure supplement(s) for figure 1:

**Figure supplement 1.** Partial protein alignment of human EPHX1 across species.

**Figure supplement 2.** Fat distribution and body composition in patient 1.

**Figure supplement 3.** Fat distribution and body composition in patient 2.

normal range: 3.6–9.6 mg/L). Mild hypertriglyceridemia was observed (2.66 mmol/L), with serum cholesterol levels around the lower limits. She had a major hepatomegaly (*Figure 1D*) and elevated levels of aspartate aminotransferase (AST – 120 IU/L), alanine aminotransferase (ALT – 148 IU/L), alkaline phosphatase (ALP – 170 IU/L), and gamma glutamyl transpeptidase (GGT – 320 IU/L) (*Table 2*). Liver computed tomography and magnetic resonance imaging (MRI) revealed liver steatosis with focal accumulation of fat depots, especially in the posterior segment (15–27%) (*Figure 1D*). Non-invasive FibroTest and Acti-test scores (*Munteanu et al., 2008*) were in favor of low-grade liver fibrosis with moderate necrosis and/or inflammation. Although pubertal development was normal, oligomenorrhea occurred rapidly and progressed over the last year to complete amenorrhea, although FSH and LH values were normal. She progressively developed hyperandrogenism signs with severe generalized hirsutism. Total serum testosterone levels were first noticed to be moderately increased at the age of 22 years (2.2 nmol/L; N: 0.3–1.5 nmol/L), with deterioration over time. At the age of 25 years, this patient presented major steroidogenesis abnormalities with especially highly elevated levels of dihydrotestosterone (2.3 nmol/L; N: 0.06–0.3 nmol/L) and testosterone (16.9 nmol/L; N: 0.3–1.5 nmol/L). Serum estradiol levels were within the normal range (121 pmol/L) in this woman with amenorrhea, contrasting with the high levels of androgens. MRI showed normal adrenal glands. Adrenal steroid profiling revealed normal levels of cortisone, cortisol, 21-desoxycortisol, 11-desoxycortisol, aldosterone, corticosterone, 21-desoxycorticosterone, 11-desoxycorticosterone, and ACTH. MRI and pelvic ultrasound did not reveal any ovarian or uterine abnormalities. Neurologically, patient 1 had a delay in language acquisition and moderate intellectual disability. An axonal neuropathy, associated with a decrease in tendon reflexes and bilateral *pes cavus*, was diagnosed at the age of 17 years and further confirmed by an electromyogram. Bilateral sensorineural hearing loss was diagnosed at the age of 10 years, leading to the use of hearing aids. Her parents were clinically unaffected.

## Clinical features in patient 2

This patient (woman) was born at term, after a normal pregnancy, with a height of 50 cm and a weight of 3.2 kg. She presented similar dysmorphic features, as compared with patient 1, including a triangular-shaped face, irregular and high hair line, frontal bossing with mid-face hypoplasia, and mandibulo-facial dysostosis (*Table 1*). Lipoatrophy was first noted in the face and the lipoatrophic phenotype was further confirmed by DXA with a total fat mass of 12.4%, a value within the first percentile as compared to age-matched normal individuals. The study of segmental body composition revealed that the loss of adipose tissue affected the whole body and was more pronounced in upper and lower limbs (*Figure 1—figure supplement 3*). Her BMI was low (17.0 kg/m$^2$). She had insulin resistance, as assessed by very high fasting insulin levels (284 pmol/L), which increased over time. Her fasting glucose values remained in the normal range, and she was not diabetic at last investigation. Measurement of serum levels of leptin (3 ng/mL) and adiponectin (0.3 mg/L) further confirmed the lipoatrophic and insulin-resistant phenotype. She had severe hypertriglyceridemia (21.9 mmol/L) associated with low HDL-cholesterol levels (0.52 mmol/L). Liver enzymes were elevated: AST (71 IU/L), ALT (94 IU/L), and ALP (145 IU/L). A liver ultrasound demonstrated fatty infiltrate. She also had profound sensorineural hearing loss since birth requiring cochlear implants. Brain computerized tomography and electroencephalogram were normal. The parents of patient 2 were clinically unaffected. Altogether, these data demonstrate that the two affected individuals have a complex disease phenotype and share a number of clinical features including dysmorphic features, lipoatrophy, insulin resistance, hypertriglyceridemia, liver dysfunction, and bilateral sensorineural hearing loss.

**Table 1.** Clinical and biological features in patients with *EPHX1* de novo variants.

Unless otherwise specified, information corresponds to that collected during the last consultation. ALP: alkaline phosphatase; ALT: alanine aminotransferase; AST: aspartate aminotransferase; DXA: dual X-ray absorptiometry; EEG: electroencephalogram; GGT: gamma glutamyl transpeptidase; MRI: magnetic resonance imaging; Na: not available; N: normal value.

|  | Patient 1 | Patient 2 |
|---|---|---|
| **General characteristics** | | |
| Origin | Sub-Saharan Africa | Western Europe |
| Age (years) | 25 | 17 |
| Sex | Female | Female |
| Height (m) | 1.62 | 1.63 |
| Weight (kg) | 53 | 45.2 |
| Body mass index (kg/m$^2$) | 20.2 | 17.0 |
| | | |
| **Birth** | | |
| At term | Yes | Yes |
| Intrauterine growth retardation | No | No |
| | | |
| **Dysmorphic features** | | |
| Microcephaly | −1.5 SD at birth −2.5 at 18 years | No |
| Triangular-shaped face | Yes | Yes |
| Irregular and high hair line | Yes | Yes |
| Frontal bossing | Yes | Yes |
| Mid face hypoplasia | No | Yes |
| Retrognathism | Yes | No |
| Mandibulo-facial dysostosis | Yes | Yes |
| Teeth misalignments | Yes | Na |
| Arachnodactyly | Yes | Na |
| | | |
| **Metabolic manifestations** | | |
| Lipoatrophy | Face, upper, and lower limbs | Face |
| Total fat mass evaluated by DXA (%) | 15.8% Z-score: −2.8 | 12.4% Z-score: −3.8 |
| Serum leptin levels (N < 28 for BMI < 25 kg/m$^2$) | 4 ng/mL | 3 ng/mL |
| Serum adiponectin levels (N: 3.6–9.6 mg/L) | 0.5 mg/L | 0.3 mg/L |
| Insulin resistance | Yes, *Acanthosis nigricans* (back, armpits, neck) Insulin requirement: up to 15 IU/kg/day before metreleptin therapy | Yes, fasting insulin: 284 pmol/L (N < 70 pmol/L) |
| Diabetes (Glycemia - N < 7 mmol/L) | Since the age of 12 Fasting glycemia: 44 mmol/L at diagnosis | No |
| Liver manifestations | Hepatomegaly, steatosis, fibrosis, liver inflammation, elevated levels of AST, ALT, ALP, and GGT | Fat infiltrate, elevated levels of AST, ALT, and GGT |
| Hypertriglyceridemia (mmol/L) (TG – N < 1.7 mmol/L) | Yes, TG: 2.66 mmol/L | Yes, TG: 21.9 mmol/L |
| Gynecological features | Clitoromegaly during childhood, oligomenorrhea | No |
| Hyperandrogenism (Testosterone – N: 0.3–1.5 nmol/L) | Generalized hirsutism, steroidogenesis alterations including high testosterone levels (16.9 mmol/L) | Na |

*Table 1 continued on next page*

*Table 1 continued*

| | Patient 1 | Patient 2 |
|---|---|---|
| Spine bone densitometry | T-score: −2.5 SD<br>Z-score: −2.5 SD | Na |
| **Neurological signs** | | |
| Bilateral sensorineural hearing loss | Since the age of 6 years and requiring hearing aids | Since birth and requiring cochlear implants |
| Developmental delay | Delay in language onset, moderate intellectual disability | No |
| Brain MRI/EEG | Normal | Normal |
| Axonal neuropathy | Since the age of 17 years<br>Decrease in osteo-tendinous reflexes (achilles, lower limbs)<br>EMG abnormalities | No |
| *Pes cavus* | Yes | Na |
| **Cardiac and musculoskeletal signs** | | |
| Cardiovascular symptoms | No | No |
| Muscular hypertrophy | Yes | No |
| Joint stiffness | Yes (hands, feet) | No |
| **Other symptoms** | | |
| Ocular signs | Bilateral cataract,<br>Peri-corneal colored ring, Diabetic retinopathy | No |
| T-cell lymphocytosis (Lymphocytes – N: 1–4.8 G/L) | Yes, 11.4 G/L<br>CD3+, CD8+, cD57+ | Na |
| Hyperkeratosis | Yes (hands, feet) | No |

## Structural characterization of *EPHX1* variants

*EPHX1* encodes a protein of 455 residues. The enzyme is retained in microsomal membranes of the ER by a single transmembrane segment located at the N-terminus and comprising around 20 amino acids (*Friedberg et al., 1994*). The C-terminal part of the protein, containing the two variants identified, is exposed at the cytosolic membrane surface and constitutes the catalytic domain (*Lewis et al., 2005*; *Figure 2A*). The EPHX1 mechanism of hydrolysis involves two chemical steps. A fast-nucleophilic attack leads to the formation of an ester intermediate, a covalent bond linking the substrate to the enzyme. Thereafter, hydrolysis of this complex to the final diol product is mediated by a molecule of water activated by a charge relay system (*Oesch et al., 2000*; *Figure 2A*). The EPHX1 active site is composed of a so-called catalytic triad consisting of Asp226, Glu404, and His431. In addition, two tyrosine residues (Tyr299 and Tyr374) provide an essential support by polarizing the epoxide (*Oesch et al., 2000*; *Bell and Kasper, 1993*; *Arand et al., 1999*). The localization of the mutated amino acids within the three-dimensional (3D) protein structure strongly supported their pathogenic effect. Although the exact structure of EPHX1 is still not available, the quaternary structure of a closely homologous enzyme was determined from the fungus *Aspergillus niger* (*Zou et al., 2000*). Glycine 430 mutated in patient 2 is located beside the crucial His431, which is directly implicated in the water activation and hydrolytic step of the catalytic process (*Figure 2B*). We used a 3D structure model from the SWISS-MODEL repository to determine the location of Thr333 (*Waterhouse et al., 2018*). On the 3D structure, this residue appears in close vicinity to Gly430, as well as to the three critical residues of the catalytic site (Asp226, Glu404, His431) (*Figure 2C*). The location of the two de novo EPHX1 variants suggests that they could affect the enzyme activity and argues for their pathogenic effect.

**Table 2.** Evolution of metabolic markers in patient 1 over a period of 6 months of metreleptin treatment.

For data before treatment, values are given as the ranges observed over the last past 3 years. AST: aspartate aminotransferase, ALT: alanine aminotransferase, ALP: alkaline phosphatase, GGT: gamma glutamyl transpeptidase.

| | Before metreleptin | After 3 month metreleptin therapy (5 mg/day) | After 6 month metreleptin therapy (7.5 mg/day) |
|---|---|---|---|
| **Anthropometric markers** | | | |
| Weight | 53 | 51 | 50 |
| BMI | 20.2 | 19.4 | 18.9 |
| **Glucose homeostasis** | | | |
| HbA1c (%) (N: 4–6%) | 11.6–16.5 | 7.9 | 7.3 |
| **Liver assessment** | | | |
| AST (IU/L) N: 17–27 IU/L | 83–120 | 57 | 54 |
| ALT (IU/L) N: 11–26 IU/L | 81–148 | 50 | 59 |
| ALP (IU/L) N: 35–105 IU/L | 100–110 | 92 | 102 |
| GGT (IU/L) N: 8–36 IU/L | 170–320 | 58 | 67 |
| Steatosis (SteatoTest) | Low-grade (S1) | Not detectable (S0) | Low grade (S1) |
| Fibrosis (FibroTest) | Intermediate grade (F1–F2) | Not detectable (F0) | Not detectable (F0) |
| Necrotic and inflammatory activity (ActiTest) | Intermediate grade (A1–A2) | very low grade (A0–A1) | Low grade (A1) |
| **Lipid profile** | | | |
| Triglycerides (mmol/L) (N: 0.4–1.7 mmol/L) | 1.3–2.7 | 2.0 | 1.6 |
| **Insulin requirement** | | | |
| Human insulin (daily doses – IU/kg) | 2.9 | 2 | 1.65 |

## *EPHX1* variants dramatically alter the enzyme hydrolysis function

To investigate the functional consequences of the identified *EPHX1* variants, we first analyzed their effect on the capacity of EPHX1 to hydrolyze *cis*-stilbene oxide ($[^3H]$-cSO), one of its well-known substrates (*Nithipatikom et al., 2014*; *Morisseau and Hammock, 2007*). Overexpression studies were performed in human epithelial kidney (HEK) 293 cells, since they are of human origin, easily transfectable, and display low endogenous levels of EPHX1. HEK 293 were transfected with plasmids encoding wild-type (WT) and mutated forms of human EPHX1 with a C-terminal Flag tag: hEPHX1[WT], hEPHX1[Thr333Pro], and hEPHX1[Gly430Arg] (*Figure 2D*). c-SO hydrolysis in lysates of HEK 293 cells overexpressing hEPHX1[WT] was high, as compared to untransfected cells in which it was undetectable. The two variants identified in patients, p.Thr333Pro and p.Gly430Arg, led to a near-complete absence of this enzyme activity (*Figure 2D,E*). We then analyzed the effect of two SNPs frequent in the general population, p.Tyr113His and p.His139Arg, whose role was debated in association studies (*El-Sherbeni and El-Kadi, 2014*; *Hassett et al., 1994*; *Figure 2B*). Neither overexpression of hEPHX1[Tyr113His] nor that of hEPHX1[His139Arg] led to reduced c-SO hydrolysis in HEK 293 lysates as compared to hEPHX1[WT] (*Figure 2D,E*). We also investigated the effect of a variant affecting one of

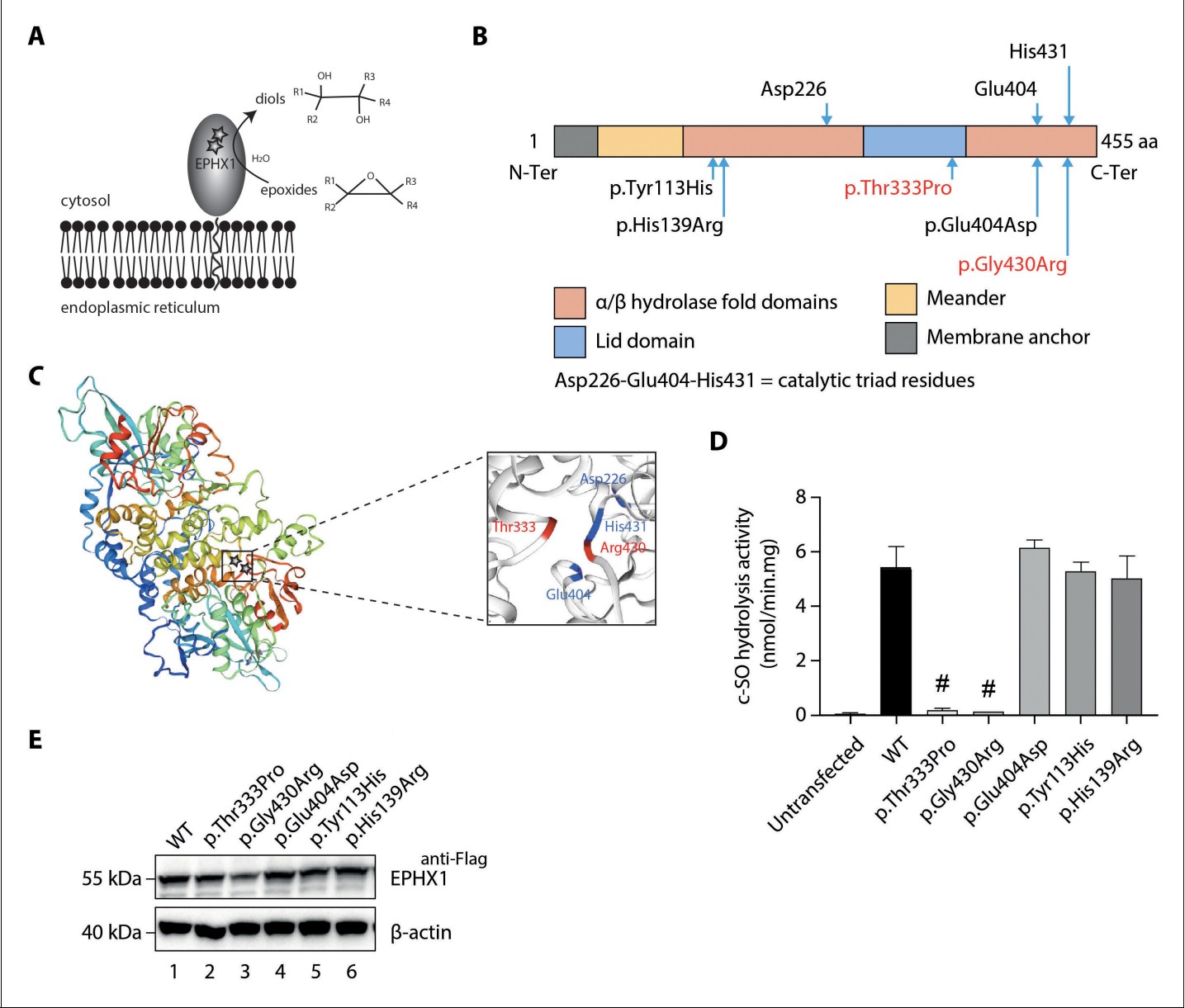

**Figure 2.** Loss of EPHX1 hydrolysis activity due to the p.Thr333Pro and p.Gly430Arg variants. (**A**) Schematic representation of the EPHX1 protein, showing its sub-cellular localization and function. Epoxide hydrolases open three membered cyclic ethers, known as epoxides, by the addition of water to yield 1,2-diols. The location of the amino acids affected by the missense variants identified in this study are indicated by stars. (**B**) Schematic representation of the variants used in functional tests. Residues of the catalytic triad are shown above the protein structure. Variants used in functional assays are depicted below. Variants identified in patients are displayed in red. (**C**) Model of the 3D structure of EPHX1, based on the quaternary structure of the closely homologous EH enzyme from the *Aspergillus niger* fungus (*Zou et al., 2000*). On the left panel, the location of the two variants identified in patients are indicated by a star. On the right panel, the two variants found in patients are depicted in red and the three key residues of the catalytic site in blue. (**D**) c-SO (*cis*-stilbene oxide) hydrolysis assay performed in HEK 293 cells transiently expressing Flag-tagged wild-type (WT) and mutated forms of human EPHX1, as indicated. Results are expressed as means ± SEM of three independent biological experiments, each of them being performed in duplicates. # indicates that hydrolysis activity of EPHX1 carrying the p.Thr333Pro and p.Gly430Arg de novo variants was abolished, compared with WT and other variants. (**E**) Western blot analysis aimed at controlling the expression of WT and mutant forms of EPHX1 in protein extracts used in c-SO hydrolysis assays presented in (**D**), using antibodies as indicated. Numbers on the left correspond to molecular weight markers (kDa). Western blot images are representative of two independent experiments.

The online version of this article includes the following figure supplement(s) for figure 2:

**Figure supplement 1.** Oxylipin profiling in patient 1.

**Figure supplement 2.** *cis*-stilbene oxide (c–SO) hydrolysis activity of co-expressed WT and mutant EPHX1 isoforms.

the catalytic triad residues, p.Glu404Asp, previously proposed to result in an increased enzyme activity (*Arand et al., 1999*; *Marowsky et al., 2016*; *Figure 2B*). This variant did not significantly modify the hydrolysis of c-SO compared to hEPHX1$^{WT}$ (*Figure 2D,E*). Immunoblot analysis against the Flag epitope was used to control the protein level of WT and mutated EPHX1 isoforms. Although the protein expression level was slightly diminished for the p.Gly430Arg variant, there was no significant difference in the expression of the WT and other mutated isoforms (*Figure 2E*). Collectively, these data showed that the p.Thr333Pro and p.Gly430Arg variants strongly impair EPHX1 hydrolysis function by altering its catalytic triad domain (*Figure 2C*).

To evaluate the impact of the loss of enzyme activity in vivo, we measured by liquid chromatography (LC) coupled with tandem mass spectrometry (MS/MS) circulating levels of a panel of EpFAs and corresponding diols in plasma samples of patient 1. EPHX1 was previously shown to catalyze the hydrolysis of several EpFAs, also called oxylipins (*Edin et al., 2018*; *Snider et al., 2007*; *Marowsky et al., 2017*; *Blum et al., 2019*). These profiles, which result from the combined action of several EHs, were compared to the patterns determined in eleven sex- and age-matched control women with normal BMI. An accumulation of 7,8 epoxydocosapentaenoic acid, and a decrease of the corresponding diol (7,8 dihydroxydocosapentaenoic acid), was observed (*Figure 2—figure supplement 1*). A recent study shows that mEH plays a significant role in the metabolism of this EpFA (*Morisseau et al., 2021*). Since oxylipin profiling is an emerging field, whose biological interpretation remains difficult, further experiments will be required to confirm this observation in additional patients and/or different cellular models.

## *EPHX1* variants induce the enzyme aggregation within the endoplasmic reticulum

As mentioned previously, EPHX1 is mainly localized in the microsomal fraction of the ER (*Coller et al., 2001*). We performed immunofluorescence staining in HEK 293 cells transiently expressing the WT and mutated forms of EPHX1 to evaluate whether missense variants alter its subcellular localization. Co-staining with calnexin, which is a marker of ER, confirmed that hEPHX1$^{WT}$ is located in the ER, as well as all mutated EPHX1 isoforms carrying the five previously mentioned missense variants (*Figure 3A*). However, the EPHX1 isoforms carrying the two de novo variants identified in patients (p.Thr333Pro and p.Gly430Arg) were also found to form higher-order complexes or clumps within the ER, as compared to WT hEPHX1 and other mutated isoforms (*Figure 3A*). To ensure that the Flag tag did not alter EPHX1 sub-cellular localization, a new set of constructs lacking the Flag tag was generated. Of note, the two de novo variants still led to EPHX1 aggregation within the ER when the Flag tag was removed (*Figure 3B*). The presence of these oligomers was further confirmed by western blot analysis since both hEPHX1$^{Thr333Pro}$ and hEPHX1$^{Gly430Asp}$ proteins were revealed as two bands, one corresponding to the protein monomer around 55 kDa, and another to an oligomer around 150 kDa (*Figure 3C*). When the cell lysates were enriched in EPHX1 by immunoprecipitation with an anti-Flag antibody, western blot analysis using an anti-hEPHX1 antibody revealed an increase of these higher-order complexes in the presence of the two p.Thr333Pro and p.Gly430Arg variants (*Figure 3C*). All these data demonstrate that the p.Thr333Pro and p.Gly430Arg variants confer an aberrant conformation to EPHX1 leading to its aggregation. This likely contributes to abolish the enzyme catalytic activity through a dominant negative mechanism.

## *Ephx1* knockout in pre-adipocytes abolishes adipocyte differentiation and decreases insulin response

We then sought to assess the effect of the loss of EPHX1 activity in the tissues most affected by the disease. The two patients have manifestations in adipose tissue, central nervous system, and liver. Recent studies have investigated the function of EPHX1 in liver and brain (*Marowsky et al., 2017*; *Marowsky et al., 2016*), but there is little information on the role of EPHX1 in adipose tissue. To investigate the role of EPHX1 in adipocytes, a CRISPR/Cas9-mediated knockout (KO) approach was developed (*Figure 4—figure supplements 1* and *2*). A custom-designed single-guide RNA (gRNA)/Cas9 expression vector targeting the sixth exon of *Ephx1* was used. The murine 3T3-L1 pre-adipocytes were chosen as a cellular model due to their ability to differentiate into mature adipocytes after stimulation in vitro (*Figure 4A*). 3T3-L1 cells transfected with a Cas9/scramble gRNA plasmid were used as a control (CTL). KO efficiency was further confirmed by western blot analysis. A major

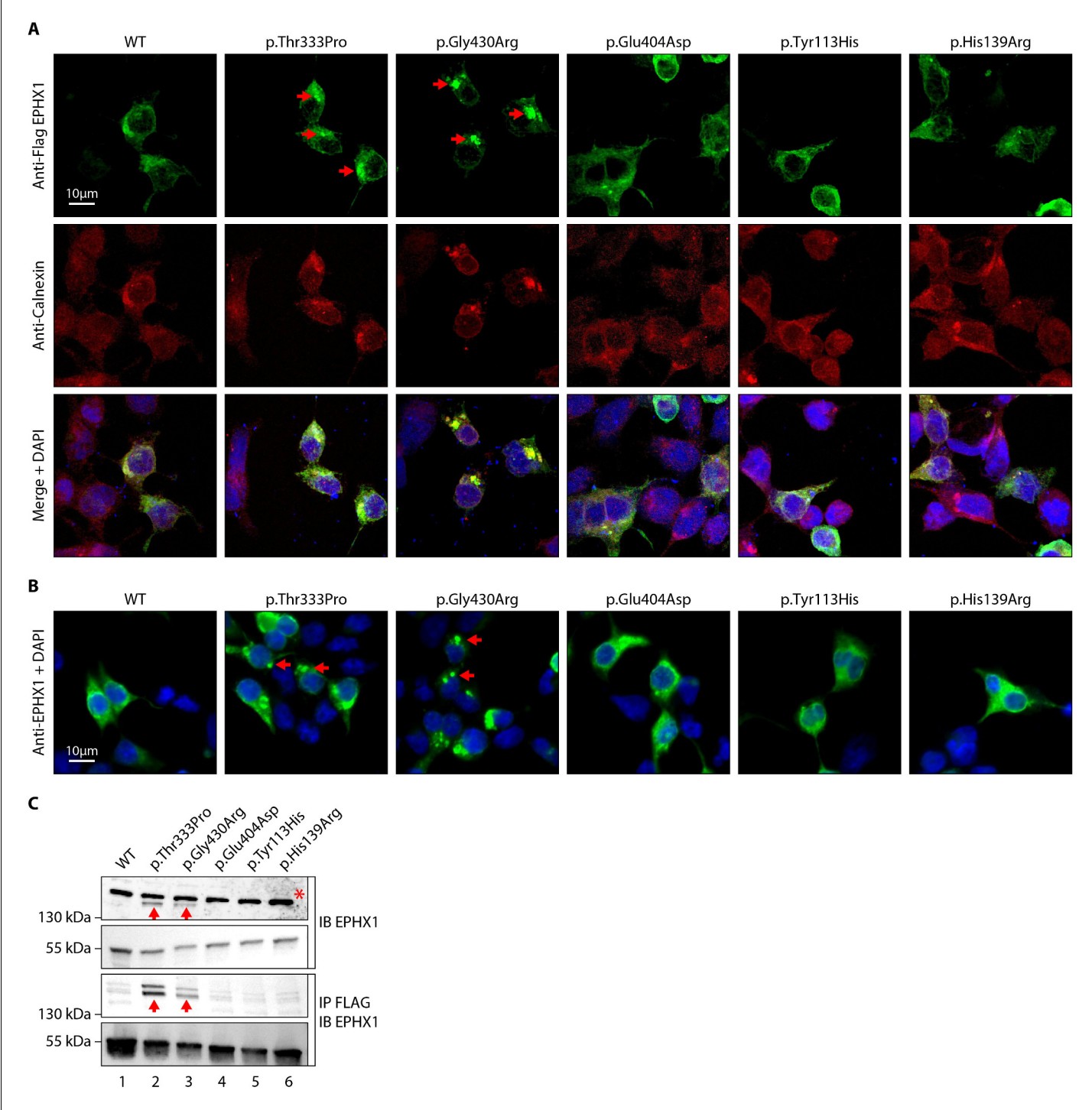

**Figure 3.** The p.Thr333Pro and p.Gly430Arg variants induce the formation of EPHX1 higher-order complexes within the endoplasmic reticulum. (**A**) HEK 293 cells transiently expressing Flag-tagged wild-type (WT) and mutated isoforms of human EPHX1 were grown on coverslips, fixed, permeabilized, and stained with an anti-Flag antibody followed by an anti-Calnexin antibody. They were then incubated with Alexa Fluor 594- and 488-conjugated secondary antibodies and visualized by confocal microscopy. Nuclei were counterstained with DAPI (blue). Red arrows point to EPHX1 higher-order complexes. Representative pictures of three independent experiments are presented. Scale bar is 10 μm. (**B**) Immunofluorescence staining of HEK 293 cells transiently expressing WT and mutated isoforms of human EPHX1 using an anti-EPHX1 antibody and visualized using an Alexa Fluor 488-conjugated goat anti-rabbit secondary antibody. Nuclei were counterstained with DAPI (blue). Representative pictures of two independent experiments are presented. Scale bar is 10 μm. (**C**) HEK 293 cells were transiently transfected with Flag-tagged WT and mutated isoforms of human EPHX1. Whole-cell extracts were prepared 24 hr later, immunoprecipitated with an anti-Flag antibody and analyzed by western blotting using an anti-EPHX1 antibody.

*Figure 3 continued on next page*

*Figure 3 continued*

The formation of EPHX1 higher-order complexes in the presence of the p.Thr333Pro and p.Gly430Arg variants is shown by red arrows. The asterisk indicates a non-specific band present only in direct immunoblotting using anti-EPHX1 antibody. Numbers on the left correspond to molecular weight markers (kDa). Western blot images are representative of three independent experiments.

loss of *Ephx1* expression, which remained stable over time during adipocyte differentiation, was indeed observed (*Figure 4B*). Consistently, hydrolysis of [³H]-cSO was evaluated in cell lysates and revealed a significant loss of enzyme activity in 3T3-L1 KO cells, as compared to control cells (*Figure 4—figure supplement 3*). Following validation of the KO model, the efficiency of adipocyte differentiation was evaluated by progressive lipid accumulation, as revealed both by the appearance of refractive droplets in optical microscopy and by an increase in Oil Red O staining, which is a marker of intracellular lipids (*Figure 4A,C*). WT and control 3T3-L1 pre-adipocytes differentiated into adipocytes within 12 days and displayed strong accumulation of lipid droplets in the cytoplasm (*Figure 4C,D*). In contrast, *Ephx1* KO led to strong and significant decrease in lipid droplet formation (p< 0.0001) (*Figure 4C,D*). The expression study of adipocyte markers constitutes another way to evaluate adipocyte differentiation. As compared to WT and control cells submitted to in vitro adipocyte differentiation, *Ephx1* KO cells displayed a significantly reduced expression of adipogenic markers, including peroxisome proliferator-activated receptor gamma (PPARγ), CCAAT/enhancer-binding protein alpha (C/EBPα), SREBP-1c, as well as reduced expression of mature adipocyte markers, such as fatty acid synthase (FAS), and adiponectin (*Figure 4E*). We next investigated the effect of the deletion of *Ephx1* on insulin sensitivity. In WT and control 3T3-L1 adipocytes stimulated with insulin, western blot analysis revealed a strong increase in the phosphorylation of several signaling intermediates from the mitotic and metabolic pathways including insulin receptor β subunit (IRβ), insulin receptor substrate-1 (IRS1), AKT, and extracellular-regulated kinase (ERK) (*Figure 4F*). In contrast, the *Ephx1* KO cells were resistant to insulin, both in pre-adipocytes and in differentiated cells, as shown by the lack or strong decrease in the phosphorylation of these intermediates upon insulin stimulation (*Figure 4F*, *Figure 4—figure supplement 4*).

To exclude the possibility that undesired off-target mutations were responsible for the effects observed in KO cells, we used another gRNA, which targets *Ephx1* exon 5. As assessed by western blot analysis, we could knock-down *Ephx1* as efficiently as with the initial gRNA (*Figure 4—figure supplement 5*). As revealed by Oil Red O staining, this new KO cellular model had a similar defect in adipocyte differentiation as the first KO cell line used throughout this study (*Figure 4—figure supplement 5*). Taken together, these results show that *Ephx1* deficiency alters adipogenesis and inhibits insulin signaling, consistent with the lipoatrophic and insulin-resistant phenotype.

### *Ephx1* knockout in pre-adipocytes promotes oxidative stress and senescence

A previous study has shown that EPHX1 might protect cells from oxidative stress (*Cheong et al., 2009*). In addition, increased cellular senescence has been functionally linked to fat-related metabolic dysfunction (*Tchkonia et al., 2010*) and has been observed in a few lipodystrophic syndromes (*Bidault et al., 2013*; *Fiorillo et al., 2018*). Cellular aging has also been associated with an increased production of reactive oxygen species (ROS) (*Davalli et al., 2016*). Consequently, we wondered if *Ephx1* deficiency might promote ROS production and senescence. To test this hypothesis, oxidative stress was first evaluated. *Ephx1* KO cells displayed higher levels of ROS in cell lysates, compared to either WT or control 3T3-L1 cells (p<0.0001) (*Figure 5A*). The proliferative capacity and biochemical markers of cellular senescence was then evaluated in edited 3T3-L1 pre-adipocytes. Bromodeoxyuridine (BrdU) incorporation was lower in *Ephx1* KO cells compared to WT and control cells (p<0.0001), consistent with a reduced proliferation rate (*Figure 5B*). In parallel, the levels of P21 and P16, two cell cycle cyclin-dependent kinase inhibitors were significantly increased in KO cells, consistent with increased senescence (*Figure 5C*). Additionally, compared to WT and control 3T3-L1 cells, *Ephx1* KO cells were characterized by a significant increase in senescence-associated β-galactosidase (SA-β-gal) activity (p<0.0001), which is another marker of cellular senescence (*Figure 5D,E*; *López-Otín et al., 2013*). Finally, enhanced levels of phosphorylated P53 were observed in KO cells (*Figure 5C*), further underlining the senescent cellular phenotype (*Qian and Chen, 2013*).

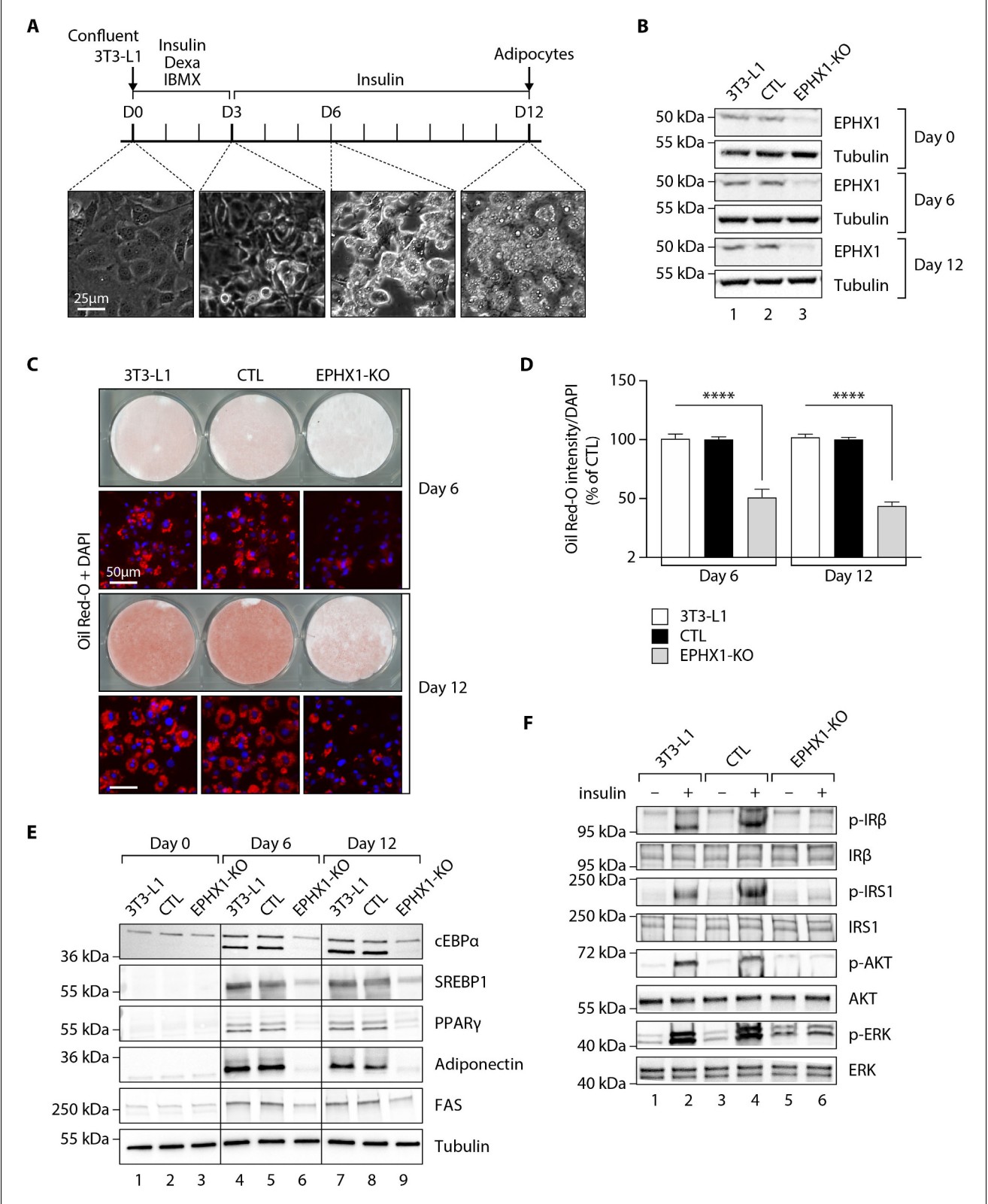

**Figure 4.** *Ephx1* deficiency suppresses adipocyte differentiation of 3T3-L1 cells and alters insulin signaling. Data were obtained in 3T3-L1 pre-adipocytes from ATCC, 3T3-L1 cells with a CRISPR-Cas9-mediated *Ephx1*-knockout (KO), and 3T3-L1 cells transfected with a Cas9/scramble gRNA plasmid corresponding to control (CTL) cells. (**A**) Timeline representation of the 3T3-L1 pre-adipocyte differentiation process using a hormonal cocktail. Dexa: dexamethasone; IBMX: 3-isobutyl-1-methylxanthine; D0–D12: Day 0 to Day 12. (**B**) Validation of *Ephx1* KO in 3T3-L1 pre-adipocytes and study of

*Figure 4 continued on next page*

*Figure 4 continued*

its expression during adipocyte differentiation. Numbers on the left correspond to molecular weight markers (kDa). Western blot images are representative of three independent experiments. (C) Adipocyte differentiation assessed by Oil Red O lipid staining. 3T3-L1 pre-adipocytes were studied during adipocyte differentiation for 12 days. First and third lines: Pictures of dishes stained by Oil Red O. Images are representative of three independent experiments. Second and fourth lines: representative images of fluorescence microscopy after staining of intracellular lipids (Oil Red O, red) and nuclei (DAPI, blue). Images are representative of five independent experiments. (D) Quantification of Oil Red O fluorescence normalized to DNA content (DAPI). Results are expressed as means ± SEM of five independent experiments. ****$p < 0.0001$. p-values were determined by analysis of variance (ANOVA) with Kruskal–Wallis post hoc multiple comparison test. (E) Protein expression of adipocyte markers obtained by western blotting during in vitro adipocyte differentiation of 3T3-L1 pre-adipocytes. Numbers on the left correspond to molecular weight markers (kDa). Western blot images are representative of three independent experiments. PPARγ: peroxisome proliferator-activated receptor gamma; C/EBPα: CCAAT/enhancer-binding protein alpha; SREBP-1c: sterol regulatory element-binding protein-1c; FAS: fatty acid synthase. (F) Activation of insulin signaling in 3T3-L1 pre-adipocytes after 10 days of adipocyte differentiation. The 3T3-L1 cells from ATCC, CTL, and *Ephx1*-KO cells were deprived of serum for 6 hr, stimulated with 20 nM insulin for 5 min or left untreated, and subjected to immunoblotting with antibodies against total and phospho-insulin receptor β-subunit (IRβ), insulin receptor substrate-1 (IRS1), AKT, and extracellular-regulated kinase (ERK)1/2. Numbers on the left correspond to molecular weight markers (kDa). Western blot images are representative of three independent experiments.

The online version of this article includes the following figure supplement(s) for figure 4:

**Figure supplement 1.** Sorting of GFP+ 3T3-L1 pre-adipocytes transfected with Cas9/scramble gRNA expression vector.
**Figure supplement 2.** Sorting of GFP+ 3T3-L1 pre-adipocytes transfected with Cas9/*Ephx1* gRNA expression vector.
**Figure supplement 3.** *cis*-stilbene oxide (c–SO) hydrolysis assay performed in 3T3-L1 cells.
**Figure supplement 4.** Activation of insulin signaling in 3T3-L1 undifferentiated pre-adipocytes.
**Figure supplement 5.** Validation of the effect of *Ephx1* KO in 3T3-L1 pre-adipocytes with a Cas9/gRNA targeting exon 5.

To further demonstrate the relevance of the 3T3-L1 murine model, a lentiviral CRISPR/Cas9-mediated *EPHX1* KO was generated in human adipose stem cells (ASCs) using a custom-designed gRNA targeting the third exon of *EPHX1*. A scramble gRNA was used as control (CTL). KO efficiency was confirmed by western blot analysis showing a near-complete loss of *EPHX1* expression (*Figure 5F*). This KO led to a major increase in cellular senescence, as assessed by the significant increase in SA-β-gal activity ($p < 0.001$) (*Figure 5G,H*) and enhanced levels of phosphorylated P53, P21, and P16 (*Figure 5F*). The level of senescence was so high (~20-fold increase) that it prevented the *EPHX1* KO ASCs to be further differentiated into adipocytes. Altogether, these data obtained in a murine cell line and validated in a human cellular model strongly argue for a functional link between EPHX1 dysfunction, oxidative stress, and cellular senescence.

## Fibroblasts from patient 1 display a senescent phenotype

Although skin is not a tissue in which *EPHX1* is highly expressed, the protein was detected by western blot in cultured fibroblasts from skin biopsies of two normal individuals (T1 and T2) and patient 1 (*Figure 6—figure supplement 1*). These immunoblot analyses using several anti-EPHX1 polyclonal antibodies allowed us to detect only the monomeric EPHX1 form (55 kDa) but not the higher-order complexes. We next assessed oxidative stress. The mutant fibroblasts showed increased levels of ROS in cell lysates compared with controls ($p < 0.0001$) (*Figure 6A*). Regarding the impact of the *EPHX1* variant on cellular senescence, the patient 1-derived fibroblasts displayed an altered morphology with an enlarged, flattened, and irregular shape, as compared to spindle-shaped control fibroblasts (*Figure 6—figure supplement 2*). BrdU incorporation was significantly reduced in mutant fibroblasts ($p < 0.0001$), which was correlated with increased levels of P21 and P16 (*Figure 6B,C*). Furthermore, SA-β-gal activity was markedly increased in the mutant fibroblasts ($p < 0.0001$), even though these fibroblasts were at an earlier passage than controls (*Figure 6D,E*). This increased SA-β-gal activity was accompanied by enhanced levels of phosphorylated P53 in mutant fibroblasts (*Figure 6C*). Together, these results were similar to those obtained using *Ephx1*-KO pre-adipocytes, confirming ex vivo the key role of EPHX1 in controlling senescence-associated oxidative stress.

## Treatment of patient 1 with metreleptin is very beneficial

Since patient 1 suffered from severe diabetes, insulin therapy was started since the age of 12 years. High doses (up to 15 IU/kg/day) of rapid acting insulin analogs and then Humulin Regular U-500 were required through a basal-bolus regimen then an insulin pump. However, this did not provide suitable glycemic control (HbA1c: 10–16.5%). Other anti-diabetic treatments, including metformin

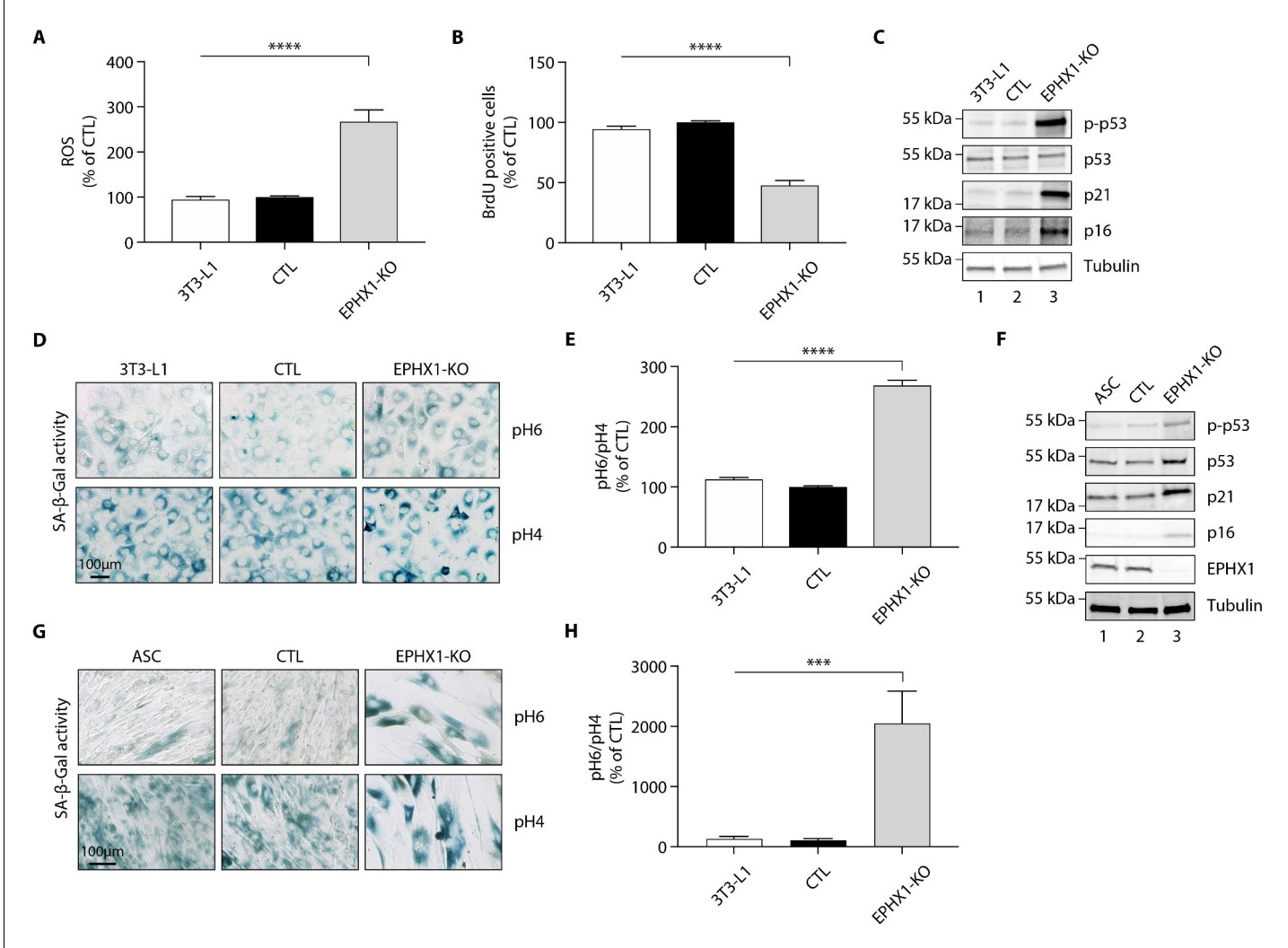

**Figure 5.** *Ephx1* deficiency causes oxidative stress and cellular senescence in murine 3T3-L1 pre-adipocytes and human ASCs. Data were obtained in 3T3-L1 pre-adipocytes from ATCC, as well as ASCs isolated from a sub-cutaneous abdominal adipose tissue biopsy from a control woman of the same sex and age as patient 1 and normal BMI. CRISPR-Cas9-mediated *EPHX1*-knockout (KO) was obtained in the two cell types. Cells transfected with a Cas9/scramble gRNA plasmid were used as control (CTL). Differences between the three cell lines were determined by analysis of variance (ANOVA) with Bonferroni's post hoc multiple comparison test. All results are expressed as means ± SEM of three independent experiments. (**A– E**) refer to 3T3-L1 cells. (**F– H**) refer to ASC cells. (**A**) Reactive oxygen species (ROS) production assessed by oxidation of 5–6-chloromethyl-2,7-dichlorodihydro-fluorescein diacetate (CM-H$_2$DCFDA) in 3T3-L1 pre-adipocytes. Results were normalized to DNA content measured by DAPI. ****p<0.0001. (**B**) Evaluation of cellular proliferation by BrdU incorporation. ****p<0.0001. (**C**) Evaluation of cellular senescence by western blotting using antibodies against the indicated proteins. Numbers on the left correspond to molecular weight markers (kDa). (**D**) Representative immunofluorescence images of senescence (SA-β-gal) after staining at pH4 and pH6. Scale bar is 100 μm. (**E**) The SA-β-gal staining ratio at pH 6.0/pH 4.0 was calculated. ****p<0.0001. (**F**) Validation of *EPHX1* KO in the ASC model and evaluation of expression of several cellular senescence markers by western blotting. Numbers on the left correspond to molecular weight markers (kDa). (**G**) Representative immunofluorescence images of senescence (SA-β-gal) after staining at pH4 and pH6. Scale bar is 100 μm. (**H**) The SA-β-gal staining ratio at pH 6.0/pH 4.0 was calculated. ***p<0.001; ****p<0.0001.

and exenatide, did not bring additional effectiveness, and the patient's hyperphagia hampered the compliance with diet. Treatment with metreleptin, a recombinant form of leptin used in the treatment of lipoatrophic syndromes, was initiated at the age of 22 years at the initial dose of 5 mg, then readjusted after 6 months at the dose of 7.5 mg. The treatment efficacy was evaluated 3 and 6 months after its introduction (*Table 2*). There was a major beneficial effect on metabolic manifestations since it led to a decrease of HbA1c to 7.3% after 6 months of treatment, allowing to reduce daily insulin doses by almost 50% (*Table 2*). Metreleptin effectiveness was also evidenced by a

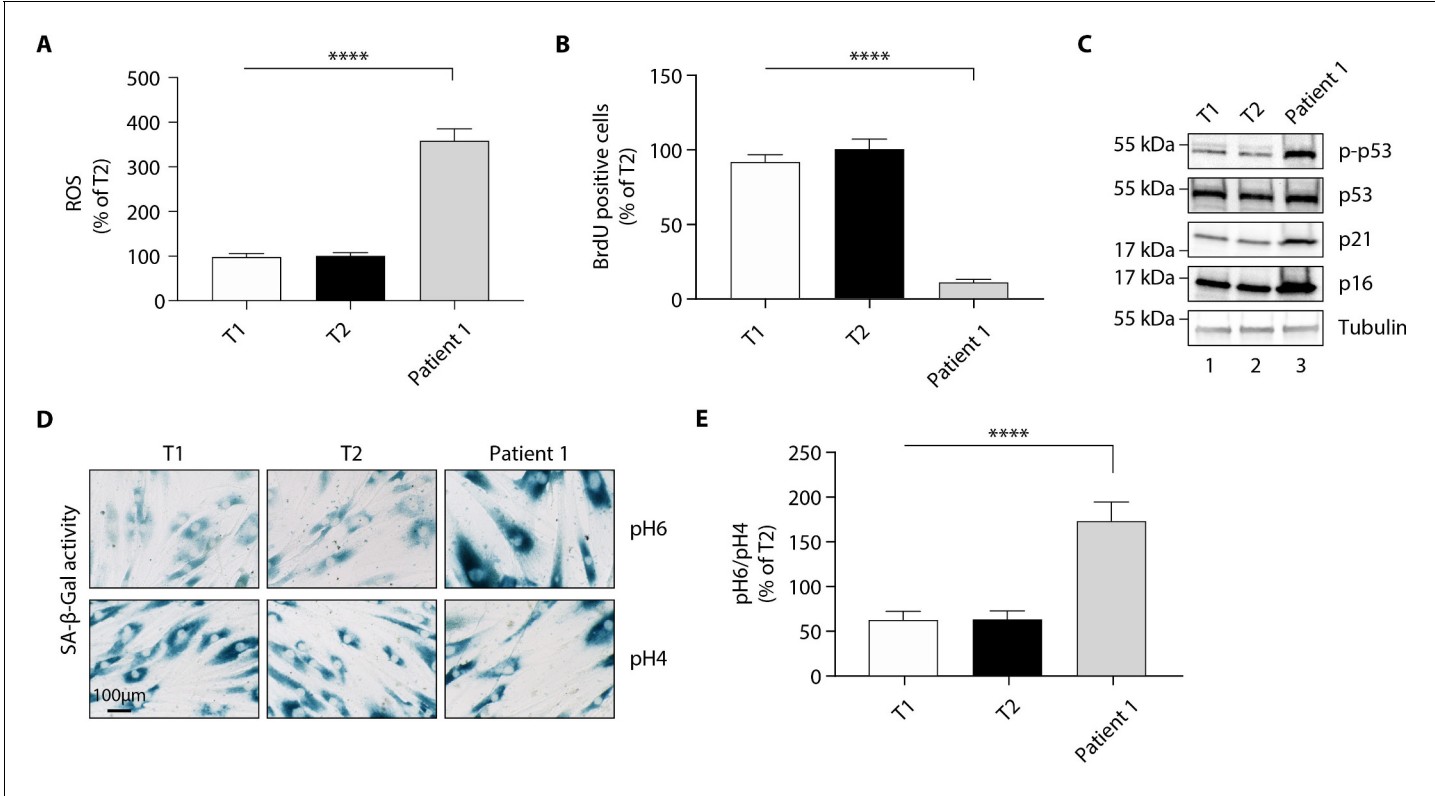

**Figure 6.** The p. Thr333Pro variant causes oxidative stress and cellular senescence in patient 1-derived fibroblasts. Data were obtained using cultured fibroblasts from skin biopsies of two normal individuals (T1 and T2) and patient 1. Differences between the three fibroblast cultures were determined by analysis of variance (ANOVA) with Bonferroni's *post hoc* multiple comparison test. All results are expressed as means ± SEM of three independent experiments. (**A**) Reactive oxygen species (ROS) production assessed by oxidation of 5–6-chloromethyl-2,7-dichlorodihydro-fluorescein diacetate (CM-H$_2$DCFDA) in fibroblasts derived from T1, T2, and patient 1. Results were normalized to DNA content measured by DAPI. ****p<0.0001. (**B**) Evaluation of cellular proliferation by BrDU incorporation. ****p<0.0001. (**C**) Evaluation of cellular senescence by western blotting using antibodies against the indicated proteins. Numbers on the left indicate molecular weight markers (kDa). (**D**) Representative immunofluorescence images of senescence (SA-β-gal) after staining at pH4 and pH6. Scale bar is 100 μm. (**E**) The SA-β-gal staining ratio at pH 6.0/pH 4.0 was calculated. ****p<0.0001.

The online version of this article includes the following figure supplement(s) for figure 6:

**Figure supplement 1.** Expression of EPHX1 in primary fibroblasts from patient 1.

**Figure supplement 2.** Comparative morphology analysis of primary fibroblasts from patient 1 and controls.

decrease in liver enzymes and an improvement of liver scores for steatosis, fibrosis, and necrosis/inflammation (*Table 2*).

## Discussion

This translational study presents the first example of a monogenic disorder related to a gene in the EH family. The data demonstrate the pleiotropic effect of EPHX1 and its key role in cell homeostasis. This is consistent with the high evolutionary conservation of EHs across multiple organisms, including animals, insects, plants, fungi, and bacteria (*van Loo et al., 2006*), and underscores their essential biological function.

In this study, two different *EPHX1* de novo missense variants were identified in patients with a progressive multisystemic disorder. According to the American College of Medical Genetics and Genomics (ACMG) criteria (*Richards et al., 2015*), these variants can be classified as 'pathogenic' with the inclusion of our functional data. The two patients share a number of clinical characteristics including dysmorphic features and manifestations affecting the liver, adipose tissue, and nervous system. The broad expression of *EPHX1* explains how germline variants in this gene result in this multisystemic phenotype. As an example, *EPHX1* transcripts have been detected in various areas of the brain (*Farin and Omiecinski, 1993*), and EPHX1 was also recently shown to play a complementary

role to EPHX2 in the brain metabolism of naturally occurring EETs, which constitute a major class of EpFAs (*Marowsky et al., 2016*; *Marowsky et al., 2009*). The involvement of the *EPHX1* de novo variants identified herein in diabetes and liver dysfunction is also consistent with a number of previous observations. Firstly, the expression of *EPHX1* is regulated by various transcriptional factors including GATA4 and HNF4A (*Liang et al., 2005*; *Peng et al., 2013*), two genes implicated in other forms of monogenic diabetes. In regard to systemic hormonal regulation, it was demonstrated in primary hepatocytes that insulin positively and glucagon negatively regulate *EPHX1* expression (*Kim et al., 2003*). A previous study also showed that a common polymorphism in *EPHX1* (p.Tyr113-His) was associated with an increased risk of type 2 diabetes mellitus and insulin resistance (*Ghattas and Amer, 2012*), as well as liver cirrhosis (*Sonzogni et al., 2002*). It was shown that EPHX1 plays a key role in the liver metabolism of endogenous lipids (*Marowsky et al., 2017*). Regarding hormonal pathophysiology, patient 1 developed amenorrhea associated with steroidogenesis alterations. Partial and generalized lipoatrophy are commonly associated with insulin resistance and hyperandrogenism (*Joy and Hegele, 2008*). Indeed, hyperinsulinemia directly stimulates ovarian androgen production, which in turn alters insulin sensitivity with a positive feedback loop between insulin resistance and hyperandrogenism. The major insulin resistance observed in patient 1 might thus contribute to hyperandrogenism signs. Nevertheless, her total testosterone levels are much higher than those usually observed in lipodystrophic patients (*Huang-Doran et al., 2021*). Such an elevation of testosterone levels in the absence of tumor and in the presence of normal estradiol levels rather argue for a direct or indirect blockade of the aromatase activity. In this regard, a potential role of EPHX1 in reproductive physiology was suggested previously. *EPHX1* is expressed in ovarian follicle cells (*Cannady et al., 2002*) and is regulated by progesterone during the menstrual cycle (*Popp et al., 2010*). Several endogenous biologically active epoxide mediators of the steroidogenic pathway were found to be EPHX1 substrates, such as androstene oxide (16α,17α-epoxyandrosten-3-one) and estroxide (epoxyestratrienol) (*Vogel-Bindel et al., 1982*). EPHX1 protects cells from oxidative stress in oviducts (*Cheong et al., 2009*). A decrease in estradiol formation from testosterone was seen in human ovaries upon treatment with an EPHX1 inhibitor (*Hattori et al., 2000*). Polycystic ovary syndrome, characterized by hyperandrogenism and elevated serum testosterone levels, is observed in patients taking sodium valproate (*Isojarvi et al., 1993*), an anti-epileptic and anti-convulsant drug known to inhibit EPHX1 activity (*Kerr et al., 1989*). This observation is reminiscent of what is seen in our patient, who displays signs of hyperandrogenism in association with a loss of EPHX1 activity. The role of EPHX1 on the reproductive function could also be illustrated by the reported link between *EPHX1* polymorphisms and spontaneous abortion (*Wang et al., 1998*) or preeclampsia (*Laasanen et al., 2002*). Consequently, further functional experiments in estrogen-producing granulosa cell models would be helpful to understand how EPHX1 could modulate aromatase activity or other steps of the steroidogenic pathway.

There were few data on the role of EPHX1 in adipose tissue, and its function in adipocytes remained elusive. The current study supports a role for EPHX1 in adipogenesis and adipocyte functions. We observed a defect in adipocyte differentiation in 3T3-L1 cells with CRISPR-Cas9 KO of *Ephx1*. This cellular model also revealed decreased expression of mature adipocyte markers, as well as altered insulin signaling even in pre-adipocytes. This is consistent with the lipoatrophic phenotype observed in the two patients carrying de novo *EPHX1* pathogenic variants. Such an adipocyte differentiation defect has been reported in several other lipoatrophic diabetes of various genetic origins (*Akinci et al., 2018a*; *Capel et al., 2018*; *Sollier et al., 2021*). Lipoatrophic diabetes are indeed characterized by an incapacity of adipose tissue to store triglycerides, leading to ectopic fat depots and insulin resistance. The profound serum leptin and adiponectin deficiency observed in patients further confirms an endocrine defect of adipose tissue, since these hormones are secreted by mature adipocytes. What is the cellular link between the loss of EPHX1 activity and adipogenesis defect? EPHX1 substrates might play a key role since oxylipins, which are EPHX1 substrates, target peroxisome proliferator-activated receptors (PPARs) to modify adipocyte formation and function (*Barquissau et al., 2017*). It has also been reported that EETs decrease mesenchymal stem cells (MSC)-derived adipocyte differentiation by inhibiting PPARγ, C/EBPα, and FAS (*Kim et al., 2010*). In addition, PPARγ agonists have been shown to increase the expression of *EPHX2* (*De Taeye et al., 2010*), whose expression is also interdependent of that of *EPHX1* (*Edin et al., 2018*). Additional experiments will be required to precisely define the link between the loss of EPHX1 and adipogenesis alteration, which might involve the deregulation of PPARγ agonists or antagonists. Altogether,

these data show that EPHX1, with its crucial role in epoxide reactive species biotransformation, stands at the crossroad of several signaling pathways and thereby plays a key role in cell metabolism and homeostasis.

EpFAs are endogenous lipid mediators functionally regulated in part by their hydrolysis by EPHX1 and EPHX2. Strategies stabilizing or mimicking EpFAs are commonly reported to contribute to cell homeostasis maintenance (*Bettaieb et al., 2013*; *Wang et al., 2014*; *Liu et al., 2018*). In this regard, it has been proposed that EpFAs prevent mitochondrial dysfunction, reduce ROS formation, and alleviate ER stress (*Inceoglu et al., 2017*). EETs exhibit numerous beneficial effects, such as anti-inflammatory, analgesic, vasodilatory, angiogenic, fibrinolytic, tissue-regenerating, and cytoprotective effects (*Marowsky et al., 2017*; *Imig, 2012*; *Morisseau, 2013*). This has been deeply investigated for EPHX2, and specific inhibitors have shown beneficial effects on a wide range of apparently unrelated conditions, including diabetes, fibrosis, chronic pain, cardiovascular, and neurodegenerative diseases (*Ghosh et al., 2020*), so that several of these EPHX2 inhibitors were tested in phase I clinical trials (*Morisseau and Hammock, 2013*; *Imig and Hammock, 2009*). In contrast, the current study shows that a loss of EPHX1 activity can be associated with a severe multisystemic phenotype. How to reconcile these apparently conflicting data? Like numerous other signaling molecules, the function of EpFAs strongly depend on the biological context (*Inceoglu et al., 2017*). The absence of a measurable effect of EpFAs under basal conditions is consistently observed across different laboratories and is in stark contrast to their strong efficacy under pathological situations. EpFAs seem to function with fine tuning to regulate and maintain cell homeostasis. This point is best illustrated by the ability of EPHX2 inhibitors to shift both hypertension and hypotension toward normotension in rodent models (*Sinal et al., 2000*; *Ulu et al., 2016*).

The gnomAD database, which collects variants from the general population, reports several dozen predicted loss-of-function variants in *EPHX1*, including nonsense, frameshift, and canonical splice site variants. This suggests that the pathogenic effect of the missense variants identified in patients 1 and 2 is not only due to a loss-of-function, but may be associated with a dominant negative mechanism. A previous study on the 3D-structure of EPHX1 suggests that this enzyme, which is mostly membrane bound, might be dimeric (*Zou et al., 2000*). This would be in favor of a dominant negative effect, as well as the formation of higher-order complexes within the ER observed in overexpression studies. Although this aspect of EPHX1 structure has so far been poorly explored, it may have important consequences for EPHX1 function, for example, by pre-orienting the enzyme for binding of either substrates or functional partners. We did not detect any dominant negative effect in the c-SO hydrolysis assay when co-expressing WT and mutated forms of EPHX1 (*Figure 2—figure supplement 2*). However, overexpression studies are probably not the most appropriate system to investigate such properties. Additional studies will be required to better understand EPHX1 activity when embedded in the microsomal ER membranes in endogenous conditions. In any case, the final result is a loss of the enzyme activity, consistent with the localization of the variants at the center of EPHX1 catalytic site. *EPHX1* KO in 3T3-L1-pre-adipocytes revealed a high rate of ROS production, an observation confirmed in patient 1-derived fibroblasts. These data reinforce the proposed role of EHs as regulators of oxidative stress (*Cheong et al., 2009*; *Inceoglu et al., 2017*; *Ulu et al., 2016*). Patient 1-derived fibroblasts displayed an altered morphology, senescent features such as an increase in the SA-β-gal marker, as well as a reduced proliferative rate. This senescent phenotype was reproduced by removing *EPHX1* in 3T3-L1 pre-adipocytes and ASCs, which subsequently loss their capacity to differentiate into adipocytes. Our findings not only support the notion that cellular senescence is an important player in adipogenesis, but also argue that EPHX1 might directly regulate adipocyte differentiation.

Regarding available animal models, *Ephx1* knockout mice have already been generated (*Miyata et al., 1999*). Deletion of the gene in the heterozygous or homozygous state did not induce an obvious phenotype and histological examination of several organs revealed no difference between *Ephx1*-null mice and wild-type littermates. The first complexity when attempting to model lipodystrophy in mice is the differences between human and murine fat distribution and lipid metabolism, particularly the handling and oxidation of lipids (*Rochford, 2014*). So far, it appears that observations made in models of congenital generalized lipodystrophies (CGL) can translate quite accurately from mice to humans. Indeed, the most frequent CGL are autosomal recessive disorders due to bi-allelic null variants. In contrast, familial partial lipodystrophy syndromes are more difficult to study in mice, since they are often autosomal dominant and some variants induce a dominant

negative effect (*Garg, 2011*). Knock-in mice would be required to better investigate such pathophysiological mechanisms for *EPHX1* variants, and additional metabolic stress is sometimes needed to uncover more aspects of the human phenotype (*Gray et al., 2006*).

In this study, we were also interested in improving the therapeutic management of this novel clinical entity, which justifies careful clinical evaluation and requires multidisciplinary care. Patient 1 suffered from very severe diabetes with persistent glycemic imbalance despite very high insulin doses. Metreleptin was shown to reduce hyperphagia leading to weight loss, to improve insulin sensitivity and secretion, to reduce hypertriglyceridemia, hyperglycemia, and fatty liver disease in many patients with lipoatrophic diabetes (*Vatier et al., 2016*; *Akinci et al., 2018b*). All these beneficial effects were rapidly observed in patient 1 after treatment initiation. Besides, regarding the specific role of EPHX1 as a xenobiotic detoxifying enzyme, another practical implication of this study for the patient is the contraindication to all drugs known to be metabolized by EPHX1, especially carbamazepine, phenobarbital, and phenytoin (*El-Sherbeni and El-Kadi, 2014*). They would not be properly metabolized by EPHX1 and eliminated by the patient leading to toxicity.

The data presented here emphasize that lipid mediator regulation by EHs is essential for homeostasis and that its alteration is a newly discovered mechanism in monogenic insulin-resistant lipoatrophic diabetes. The field of monogenic diabetes is quickly advancing, and knowledge gained in recent years led to great improvements in understanding of their molecular and cellular bases. This allows to improve overall quality of life for patients and their families, with earlier diagnoses and personalized treatments. Continued efforts at gene discovery may help reveal mechanistic pathways implicated in the more common forms of type 1 and type 2 diabetes and lead to better treatments and outcomes.

# Materials and methods

## Key resources table

| Reagent type (species) or resource | Designation | Source or reference | Identifiers | Additional information |
|---|---|---|---|---|
| Cell line (*Homo sapiens*) | HEK 293 | ATCC | CRL-1573 | Embryonic kidney |
| Cell line (*Mus musculus*) | 3T3-L1 | ATCC | CL-173 | The cells undergo a pre-adipose to adipose like conversion as they progress from a rapidly dividing to a confluent state |
| Primary fibroblasts | T1 | Pr. Fève lab at CRSA, Paris | N/A | Non-obese and non-diabetic female, skin biopsy |
| Primary fibroblasts | T2 | Pr. Fève lab at CRSA, Paris | N/A | Non-obese and non-diabetic female, skin biopsy |
| Primary fibroblasts | Patient 1 | Pr. Fève lab at CRSA, Paris | N/A | Patient 1, female, skin biopsy |
| Adipose stem cells | ASCs | Pr. Fève lab at CRSA, Paris | N/A | Female, from subcutaneous abdominal adipose tissue |
| Antibody | Anti-adiponectin | Thermo Fisher Scientific | Cat# MA1-054 | WB (1:1000) |
| Antibody | Anti-AKT | Santa Cruz Biotechnology | Cat# sc-8312 | WB (1:1000) |
| Antibody | Anti β-actin | Sigma Aldrich | Cat# A2228 | WB (1:10,000) |
| Antibody | Anti-Calnexin | ENZO Life Science | Cat# ADO-SPA-860 | IF (1:200) |
| Antibody | Anti-C/EPBα | Protein Tech | Cat# 18311-1-1P | WB (1:1000) |
| Antibody | Anti-EPHX1 | Novus | Cat# NBP1-3301 | WB (1:1000) - IF (1:1000) |
| Antibody | Anti-ERK | Cell Signaling Technology | Cat# 9102 | WB (1:1000) |
| Antibody | Anti-FAS | Cell Signaling Technology | Cat# 3180 | WB (1:1000) |
| Antibody | Anti-Flag | Origene | Cat# TA50011-100 | WB (1:1000) - IF (1:1000) - IP (1:200) |
| Antibody | Anti-IRB | Cell Signaling Technology | Cat# 3025 | WB (1:1000) |
| Antibody | Anti-IRS1 | Protein Tech | Cat# 17509–1-AP | WB (1:1000) |

*Continued on next page*

*Continued*

| Reagent type (species) or resource | Designation | Source or reference | Identifiers | Additional information |
|---|---|---|---|---|
| Antibody | Anti-P16 | Protein Tech | Cat# 10883–1-AP | WB (1:1000) |
| Antibody | Anti-P21 | Protein Tech | Cat# 10355–1-AP | WB (1:1000) |
| Antibody | Anti-P53 | Abcam | Cat# ab1101 | WB (1:1000) |
| Antibody | Anti-P-AKT | Santa Cruz Biotechnology | Cat# sc-7985-R | WB (1:1000) |
| Antibody | Anti-P-ERK | Cell Signaling Technology | Cat# 9101 | WB (1:1000) |
| Antibody | Anti-P-P53 | Abcam | Cat# ab38497 | WB (1:1000) |
| Antibody | Anti-PPARg | Protein Tech | Cat# 16643–1-AP | WB (1:1000) |
| Antibody | Anti-SREBP-1 | Santa Cruz Biotechnology | Cat# sc-366 | WB (1:1000) |
| Antibody | Anti-Tubulin | Protein Tech | Cat# 66031–1-lg | WB (1:10,000) |
| Antibody | Anti-P-Tyr | Santa Cruz Biotechnology | Cat# sc-7020 | WB (1:1000) |
| Antibody | Anti-rabbit-HRP | GE Healthcare | Cat# NA934V | WB (1:2000) |
| Antibody | Anti-mouse-HRP | GE Healthcare | Cat# NA931V | WB (1:2000) |
| Recombinant DNA reagent (plasmid) | pCMV-entry-Flag | Origene | Cat # PS100001 | |
| Recombinant DNA reagent (plasmid) | pCMV-EPHX1 WT-Flag or without Flag | This paper | N/A | Described in Materials and methods Publicly available (Addgene) |
| Recombinant DNA reagent (plasmid) | pCMV-EPHX1 c.337T>C - Flag or without Flag | This paper | N/A | Described in Materials and methods Publicly available (Addgene) |
| Recombinant DNA reagent (plasmid) | pCMV-EPHX1 c.416A>G - Flag or without Flag | This paper | N/A | Described in Materials and methods Publicly available (Addgene) |
| Recombinant DNA reagent (plasmid) | pCMV-EPHX1 c.997A>C - Flag or without Flag | This paper | N/A | Described in Materials and methods Publicly available (Addgene) |
| Recombinant DNA reagent (plasmid) | pCMV-EPHX1 c.1212G>C - Flag or without Flag | This paper | N/A | Described in Materials and methods Publicly available (Addgene) |
| Recombinant DNA reagent (plasmid) | pCMV-EPHX1 c.1288G>C - Flag or without Flag | This paper | N/A | Described in Materials and methods Publicly available (Addgene) |
| Recombinant DNA reagent (plasmid) | pSpCas9(BB)—2A-GFP (PX458) | Addgene | Cat# 48138 | A gift from Zhang lab |
| Recombinant DNA reagent (plasmid) | pLentiCRISPR v2 | Addgene | Cat# 52961 | A gift from Zhang lab |
| Software algorithm | FIJI software | NIH | N/A | |
| Software algorithm | GraphPad | Graphpad Software | N/A | |
| Software algorithm | Prism | Graphpad Software | N/A | |

## Genetic studies

Diagnostic laboratories performed genetic analyses on genomic blood DNA extracted from peripheral blood leukocytes using standard procedures. Exons and flanking intronic sequences of a panel of genes involved in lipoatrophic diabetes (*Jéru et al., 2019*) were captured from fragmented DNA with the SeqCapEZ enrichment protocol (Roche NimbleGen, USA). Paired-end massively parallel sequencing was achieved on a MiSeq platform (Illumina, USA). Bioinformatic analysis was performed using the Sophia DDM pipeline (Sophia Genetics, Switzerland). Identification of *EPHX1* variants was obtained by WES. For patient 1, library preparation, exome capture, sequencing, and variant calling and annotation were performed by IntegraGen SA (Evry, France). Genomic DNA was captured using Twist Human Core Exome Enrichment System (Twist Bioscience, USA) and IntegraGen Custom, followed by paired-end 75 bases massively parallel sequencing on Illumina HiSeq4000. Identification of

the potentially pathogenic variants were determined using Sirius software (IntegraGen SA, France). For patient 2, exome sequencing was performed on the proband and both parents as previously described (*Retterer et al., 2016*). *EPHX1* variants were confirmed by Sanger sequencing with the Big Dye Terminator v3.1 sequencing kit (Thermo Fisher Scientific, MS, USA) after PCR amplification and analyzed on a 3500xL Dx device with the SeqScape v2.7 software (Thermo Fisher Scientific, MS, USA). *EPHX1* variants were described based on the longest isoform (NM_000120.4) using Alamut 2.11 (Sophia Genetics, Switzerland) and Human Genome Variation Society guidelines. Variants were classified according to the ACMG recommendations (*van Loo et al., 2006*). Protein sequences were aligned using the Clustal Omega software, and the degree of conservation was presented with help of the BoxShade software. *EPHX1* variants were queried in human populations using gnomAD.

## Plasmids and transfection

*EPHX1* cDNA was amplified by RT-qPCR using RNA from HepG2 cells and inserted into the pCMV6-entry mammalian expression vector containing a C-terminal Flag Tag (#PS100001; Origene, MD, USA). *EPHX1* variants (c.337T>C, c.416A>G, c.997A>C, c.1212G>C, and c.1288G>C) were introduced using the Quikchange II Site-directed mutagenesis kit (#200523; Agilent Technologies, CA, USA), and constructs were checked by Sanger sequencing. To remove the C-terminal Flag Tag of the different plasmids, a nonsense variant was introduced by site-directed mutagenesis. Transient transfection of the different cell lines was carried out in six-well plates with TurboFect Transfection Reagent (#R0532; Thermo Fisher Scientific, MS, USA) according to the manufacturer's instructions.

## Cell culture

HEK 293 cells purchased from ATCC with a negative mycoplasma contamination test were cultured in high-glucose (4.5 g/L) Dulbecco's modified Eagle's medium (DMEM; #11960085; Thermo Fisher Scientific) containing 10% fetal calf serum (FCS) (#F7524; Sigma-Aldrich, MI, USA), 1% penicillin/streptomycin (PS). 3T3-L1 pre-adipocytes purchased from ATCC with a negative mycoplasma contamination test were maintained in an undifferentiated state in high-glucose (4.5 g/L) DMEM supplemented with 10% newborn calf serum and 1% PS (#CA-1151500; Biosera, MI, USA). Adipocyte differentiation was induced by treating 2 day post-confluent cultures with high-glucose (4.5 g/L) DMEM supplemented with 10% FCS, 1% PS, 1 µM dexamethasone (#D4902; Sigma-Aldrich), 250 µM 3-isobutyl-1-methyl xanthine (IBMX) (#I7018; Sigma-Aldrich), and 0.17 µM insulin (#I0516; Sigma-Aldrich) for 3 days. The medium was then replaced with high-glucose DMEM supplemented with 10% FCS, 1% PS, and 0.17 µM insulin and changed to fresh medium every 2 days until the 12th day. Primary fibroblast cultures were established using skin biopsies from two healthy non-obese non-diabetic women, named T1 and T2, as well as from patient 1. Fibroblasts were grown in low glucose (1 g/L) DMEM with pyruvate (#31885049; Thermo Fisher Scientific) and supplemented with 10% FCS, 1% PS, and 2 mM glutamine. Fibroblasts were stained at a low passage number (i.e., passage 4 for patient 1, passage 9 for T1 and T2). ASCs were isolated from surgical samples of sub-cutaneous abdominal adipose tissue from a control woman of the same sex and age as patient 1 and normal BMI. Adipose tissue samples were enzymatically digested with collagenase B (0.2%). After centrifugation, stromal vascular fraction was filtered, rinsed, plated, and cultured in α-MEM with 10% FCS, 2 mmol/L glutamin, 1% PS (10,000 UI/mL), 1% HEPES, and fibroblast growth factor-2 (145 nmol/L). After 24 hr, only ASCs adhered to plastic surfaces, while other cells were removed after culture medium replacement. ASCs were maintained in an undifferentiated state in high-glucose (4.5 g/L) DMEM supplemented with 10% newborn calf serum and 1% PS. All culture conditions were kept constant throughout the experiments.

## CRISPR/Cas9-mediated deletion of *Ephx1* in 3T3-L1 pre-adipocytes

pSpCas9(BB)—2A-GFP (PX458) was a gift from Zhang lab (Addgene, MA, USA; plasmid #48138) and was used to transfect 3T3-L1 cells with Cas9 along with the targeting guide RNAs (gRNAs). gRNAs were designed and checked for efficiency (http://cistrome.org/SSC) and specificity (http://crispr.mit.edu). We used the web-based tool, CRISPOR (http://crispor.tefor.net/), to avoid off-target sequences. Subsequently, gRNAs were cloned in the plasmid and transfected into cells using TurboFect (#R0532; Thermo Fisher Scientific) according to the manufacturer's instructions. Forty-eight hours post-transfection, cells were sorted by flow cytometry (Cell Sorting Core Facility, Centre de

Recherche Saint-Antoine), and cells with the highest GFP positivity were transferred into a 24-well plate and propagated. We favored a plasmid transient transfection method rather than a viral transduction to reduce the expression of Cas9 in cells and prevent its integration into the host cell genome, which may lead to increased off-target activities. Moreover, to minimize the effect of possible off-target mutations, we analyzed heterogeneous populations issued from the FACS sorting rather than clonal populations. The gRNA sequences used in this study to target *Ephx1* were the following: gRNA (exon 6) sense:

5' TCTTAGAGAAGTTCTCCACC 3'; antisense: 5' GGTGGAGAACTTCTCTAAGA 3'. gRNA (exon 5) sense: 5' TACAACATCATGAGGGAGAG 3'; antisense: 5' CTCTCCCTCATGATGTTGTA 3'.

## CRISPR/Cas9-mediated deletion of *EPHX1* in human ASCs

The lentiviral plasmid plentiCRISPRv2 was a gift from Zhang lab (Addgene, MA, USA; plasmid #52961) and contains the puromycin resistance, hSpCas9 and the chimeric guide RNA (gRNAs). The gRNA targeting *EPHX1* exon 3 was designed and checked for efficiency (http://cistrome.org/SSC) and specificity (http://crispr.mit.edu). Its sequence was the following: sense 5' CCCTGGCTATGGC TTCTCAG 3'; antisense 5' CTGAGAAGCCATAGCCAGGG 3'. The web-based tool, CRISPOR (http://crispor.tefor.net), was used to evaluate potential off-target sequences. Subsequently, the gRNA was cloned into plentiCRISPRv2 and lentivirus were produced by the VVTG platform (SFR Necker, France). ASCs were infected with viral particles at a minimal titer of $10^8$ transducing units per mL. Forty-eight hours post-infection, cells were selected with 5 μg/mL puromycin dihydrochloride (#P9620; Sigma-Aldrich). Surviving cells were propagated, and the heterogeneous cell pool was used for experiments.

## Immunofluorescence

For indirect immunofluorescence, HEK 293 cells were grown on glass coverslips, and after transfection, they were fixed for 15 min in 4% paraformaldehyde (PFA) in phosphate-buffered saline (PBS) and then permeabilized for 15 min with 0.1% Triton X-100 in PBS at room temperature. Cells were washed three times with a blocking solution containing PBS with 5% fatty acid free bovine serum albumin. Primary antibodies used for immunostaining were as follows: mouse anti-Flag (Origene) (1/ 1000), rabbit anti-EPHX1 (Novus) (1/1000), and rabbit anti-Calnexin (#ADO-SPA-860; Enzo Life Sciences, France) (1/200). Cells were incubated 1 hr at 37°C, rinsed, and then incubated for 45 min at room temperature with the appropriate Alexa-conjugated isotype-specific secondary antibodies (Thermo Fisher Scientific) and 4',6-diamidino-2-phenylindole (DAPI) (Thermo Fisher Scientific). The coverslips were mounted in DAKO fluorescence mounting media (#S3023; Agilent Technologies, CA, USA). Images were acquired using a SP2-inverted confocal microscope (Leica Biosystems, Germany), equipped with an HCX PL APO CS 63X/1.32 oil immersion objective, and analyzed using Leica Confocal Software and FIJI Software. For each experiment, all images were acquired with constant settings (acquisition time and correction of signal intensities).

## Measurement of EPHX1 activity

EPHX1 activity in lysates of transfected HEK 293 or 3T3-L1 cells was determined using [$^3$H]-*cis*-stilbene oxide ([$^3$H]-cSO), as described previously (*Hassett et al., 1994*). After thawing on ice, cells were diluted with chilled sodium phosphate buffer (20 mM, pH 7.4) containing 5 mM ethylenediaminetetraacetic acid (EDTA), 1 mM dithiothreitol (DTT), and 1 mM phenylmethylsulfonyl fluoride, and 0.2% (v/v) Triton X-100. Cells were then broken with a 10 s ultrasonic pulse. The mixture was centrifuged at 5000 g for 20 min at 4°C. Supernatants were collected and flashed frozen, before being used for further analysis. Protein concentration was quantified using the Pierce BCA assay (Pierce, IL, USA), using Fraction V bovine serum albumin (BSA) as the calibrating standard. After thawing on ice, supernatants were diluted (5- to 20-folds) with Tris–HCl buffer (0.1 M, pH 9.0) containing freshly added BSA (0.1 mg/mL). In glass tubes containing 100 μL of the diluted extract, the reaction was started by adding 1 μL of 5 mM [$^3$H]-cSO in ethanol (10,000 cpm, [S]final = 50 μM). The mixture was incubated at 37°C for 5–120 min. The reaction was then quenched by the addition of 250 μL of isooctane, which extracts the remaining epoxide from the aqueous phase. The activity was followed by measuring the quantity of radioactive diols formed in the aqueous phase using a scintillation counter (TriCarb 2810 TR, Perkin Elmer, Shelton, CT). Assays were performed in triplicates.

## Western blot

Cells were homogenized in NP-40 lysis buffer to obtain protein lysates. Thirty micrograms of protein extracts was separated by sodium dodecyl sulfate (SDS)–polyacrylamide gel electrophoresis, transferred to polyvinylidene difluoride membrane, and analyzed by immunoblotting.

## Immunoprecipitation

Transfected HEK 293 cells were recovered in 1 mL of PBS. They were centrifuged at 4025 g for 1 min in a microfuge and resuspended in 50–100 µL of TNT buffer (20 mM Tris–HCl - pH 7.5, 200 mM NaCl, 1% Triton X-100). After incubation on ice for 10 min and centrifugation at 16,099 g for 10 min, supernatants were recovered and protein concentrations determined. Protein lysates (100–200 mg in 200 µL of TNT) were either directly analyzed by western blotting or first immunoprecipitated. In this latter case, extracts were incubated under rotation for 2 hr at 4°C with the relevant antibody. Protein G Sepharose (Sigma) was then added and the mixture incubated for a further 1 hr at 4°C. Sepharose beads were quickly centrifuged in a microfuge (30 s at 11,180 g) and washed three times with TNT. After final wash, the beads were resuspended in 30 mL of buffer A (10 mM KCl, 2 mM MgCl$_2$, 0.1 mM EDTA, 1 mM DTT, 10 mM HEPES – pH 7.8) complemented with loading dye, before being processed as described in western blot analysis.

## Oxylipin extraction and quantification in plasma

The targeted oxylipin analysis was designed based on the metabolic pathway of n-3 and n-6 polyunsaturated fatty acid precursors as described previously (*Yang et al., 2009*). The LC/MS-MS analysis was performed with an Agilent 1200SL UHPLC system interfaced with a 4000 QTRAP mass spectrometer (Sciex). Separation of oxylipins was performed with the Agilent Eclipse Plus C18 150 × 2.1 mm 1.8 µm column with mobile phases of water with 0.1% acetic acid as mobile phase A and acetonitrile/methanol (84/16) with 0.1% acetic acid as mobile phase B.

## Oil Red O staining, image processing, and quantification

Intracellular lipids were stained by Oil Red O (#O0625; Sigma-Aldrich). Cells were washed with PBS and fixed with 4% PFA in PBS, for 10 min. Fixed cells were incubated with Oil Red O solution for 1 hr at room temperature and then with DAPI (Thermo Fischer Scientific) for 5 min. Fluorescence images were generated with IX83 Olympus microscope, acquired with Cell-Sens V1.6 and analyzed with FIJI software. Images of 8–10 different areas per condition were visualized by fluorescence microscopy using mCherry filter, followed by computer image analysis using FIJI software. Briefly, analysis was performed by threshold converting the 8-bit Red-Green-Blue image into a binary image, which consists only of pixels representing lipid droplets (i.e., red). Importantly, after separation, the binary image was manually compared with the original image for consistency and correct binary conversion. The area occupied by lipid droplets in the image was displayed by FIJI software as surface area in µm$^2$ and normalized to cell number by semi-automated counting of DAPI-stained nuclei.

## Cell proliferation assay

3T3-L1 cells and fibroblasts were seeded in a 12-well plate (5000 per well) and incubated overnight at 37°C in DMEM supplemented with 10% FCS and 1% PS. Cell proliferation was evaluated by BrdU incorporation using a colorimetric ELISA assay (#QIA58; Sigma-Aldrich) 16 hr after seeding, according to the manufacturer's instructions.

## Oxidative stress and cellular senescence

The oxidation of the fluorogenic probe 2,7-dichlorodihydro-fluorescein diacetate (CM-H$_2$DCFHDA) (2 µg/mL, #C6827; Sigma-Aldrich) was used to evaluate intracellular levels of ROS on a 200-plate fluorescence reader (Tecan Infinite, Switzerland) at 520–595 nm. The blue staining of β-galactosidase (β-gal) at pH 6 was used as a biomarker of cellular senescence. Cells were fixed with 4% PFA in PBS for 5 min at room temperature. Cells were washed twice with PBS and incubated overnight in fresh SA-β-gal staining solution containing 1 mg/mL of X-gal (5-bromo-4-chloro-3-indolyl-β-D-galactopyranoside) (#3117073001; Sigma-Aldrich), 5 mM potassium ferrocyanide, 5 mM potassium ferricyanide, 150 mM NaCl, 2 mM MgCl$_2$, and 0.4 mM phosphate buffer, pH 6.0, in darkness at 37°C without CO$_2$. For positive staining controls, fixed cells were treated with the same solution, but at pH 4.0.

After imaging with an IX83 Olympus microscope, stained cells were resuspended with 2% SDS, scratched, and sonicated. Finally, the absorbance (630 nm) was read with a Tecan Infinite 200-plate reader and the staining ratio at pH 6.0/pH 4.0 was calculated.

## Statistics

Data are presented as means ± SEM (standard error of the mean). GraphPad Prism software (Graph-Pad Software) was used to calculate statistical significance. Gaussian distribution was tested with the Kolmogorov–Smirnov test. Multiple comparisons were conducted by one-way analysis of variance (ANOVA) with Bonferroni test or Kruskal–Wallis test for post hoc analysis. $p < 0.05$ was considered statistically significant.

## Study approval

We obtained written informed consent for all genetic studies as well as for the use of photographs shown in *Figure 1*. The study was approved by the CPP Ile de France five research ethics board (DC 2009–963, Paris, France) and the Columbia institutional review board (AAAJ8651, New York, United States).

# Acknowledgements

We thank the patients and their families for their participation. The authors would like to thank Dr. Boris Keren (Unité Fonctionnelle de Génomique du Développement, AP-HP, Paris, France) for analyses of SNP DNA chips, Beth Hudson (GeneDx, Gaithersburg, United States) for her involvement in searching patients with *EPHX1* variants, Annie Munier (Cytométrie et Imagerie Saint-Antoine, Sorbonne Université, Paris, France) for cell sorting of transfected 3T3-L1 pre-adipocytes, Romain Morichon for image processing (UMS30 Lumic, Sorbonne Université, Paris, France), and Yves Chrétien (CRSA, Paris, France) for expert artwork.

# Additional information

## Funding

| Funder | Grant reference number | Author |
| --- | --- | --- |
| Mairie de Paris | R18139DD | Jeremie Gautheron |
| Société Francophone du Diabète | R19114DD | Jeremie Gautheron |
| Fondation pour la Recherche Médicale | ARF20170938613 | Jeremie Gautheron |
| Fondation pour la Recherche Médicale | EQU202003010517 | Jeremie Gautheron |
| Fondation pour la Recherche Médicale | EQU201903007868 | Bruno Fève Corinne Vigouroux Isabelle Jeru |
| National Institutes of Health | DK52431 | Wendy K Chung |
| National Institute of Environmental Health Sciences | R35ES030443 | Christophe Morisseau Bruce D Hammock |
| National Institute of Environmental Health Sciences | P42ES004699 | Christophe Morisseau Bruce D Hammock |

The funders had no role in study design, data collection and interpretation, or the decision to submit the work for publication.

## Author contributions

Jeremie Gautheron, Isabelle Jeru, Conceptualization, Data curation, Formal analysis, Supervision, Funding acquisition, Validation, Investigation, Methodology, Writing - original draft, Project administration, Writing - review and editing; Christophe Morisseau, Wendy K Chung, Data curation, Formal

analysis, Funding acquisition, Validation, Investigation, Writing - review and editing; Jamila Zammouri, Genevieve Baujat, Emilie Capel, Celia Moulin, Yuxin Wang, Jun Yang, Data curation, Formal analysis, Investigation, Writing - review and editing; Martine Auclair, Data curation, Formal analysis, Investigation, Methodology, Writing - review and editing; Bruce D Hammock, Barbara Cerame, Franck Phan, Data curation, Formal analysis, Writing - review and editing; Bruno Fève, Corinne Vigouroux, Fabrizio Andreelli, Conceptualization, Formal analysis, Validation, Investigation, Writing - review and editing

### Author ORCIDs

Jeremie Gautheron https://orcid.org/0000-0002-7727-9893
Bruce D Hammock https://orcid.org/0000-0003-1408-8317
Isabelle Jeru https://orcid.org/0000-0001-7171-0577

### Ethics

Human subjects: We obtained written informed consent for all genetic studies as well as for the use of photographs. The study was approved by the CPP Ile de France 5 research ethics board (DC 2009-963, Paris, France) and the Columbia institutional review board (AAAJ8651, New York, United States).

### Decision letter and Author response

Decision letter https://doi.org/10.7554/eLife.68445.sa1
Author response https://doi.org/10.7554/eLife.68445.sa2

---

## Additional files

### Supplementary files

- Source data 1. Uncropped and marked western blots seen in the different figures.

- Source data 2. Uncropped and unedited western blots seen in the different figures.

- Supplementary file 1. Characteristics of additional rare variants identified in patient 1 according to their mode of inheritance. Variants with an allele frequency > 0.001 were not considered since their frequency is not compatible with the prevalence of this very rare disorder. The gnomAD database (https://gnomad.broadinstitute.org/) was used to determine the variant frequency in the general population. Expression profile were extracted from the GTex portal (https://www.gtexportal.org). Splice variants located outside of the canonical sites were not considered. The CADD (Combined Annotation Dependent Depletion) score was used for first scoring of variant deleteriousness (https://cadd-staging.kircherlab.bihealth.org/). A score > 20 argues for a pathogenic effect. Other bioinformatic tools of pathogenicity prediction were considered including SIFT, Polyphen2, MutationTaster, and REVEL. On each line, the arguments against the involvement of a given variant in the observed phenotype are indicated by an asterisk. Each variant can be excluded by at least two independent items. hmz: homozygous; htz: heterozygous; na: not applicable.

- Supplementary file 2. Characteristics of additional rare variants identified in patient 2 according to their mode of inheritance. Variants with an allele frequency > 0.001 were not considered since their frequency is not compatible with the prevalence of this very rare disorder. The gnomAD database (https://gnomad.broadinstitute.org/) was used to determine the variant frequency in the general population. Expression profile were extracted from the GTex portal (https://www.gtexportal.org). Splice variants located outside of the canonical sites were not considered. The CADD (Combined Annotation Dependent Depletion) score was used for first scoring of variant deleteriousness (https://cadd-staging.kircherlab.bihealth.org/). A score > 20 argues for a pathogenic effect. Other bioinformatic tools of pathogenicity prediction were considered including SIFT, Polyphen2, MutationTaster, and REVEL. On each line, the arguments against the involvement of a given variant in the observed phenotype are indicated by an asterisk. Each variant can be excluded by at least two independent items. hmz: homozygous; htz: heterozygous; na: not applicable.

- Transparent reporting form

## Data availability

Plasmids used in this study have been deposited in Addgene under accession number 79368. Exome data from human subjects cannot been made available, since the informed written consents did not permit sharing of the full sequence data. To circumvent this fact and to fit with the Editor and Reviewers' comment, two supplementary files have been provided listing all other rare variants identified in the two families investigated herein, according to the corresponding mode of inheritance. For each variant, we indicated the items arguing against its involvement in the disease phenotype. All data generated or analysed during this study are included in the manuscript and supporting files. Source data have been provided for all western blot experiments.

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
