## [Decision Letter]

**Acceptance summary:**

This manuscript describes a novel genetic basis for human lipodystrophy associated with diabetes and insulin resistance. Two human subjects with mutations in the catalytic region of the epoxide hydrolase EPHX1 which catalyzes hydrolysis of epoxides to diols display this disease, and disruption of this enzyme in cultured cells attenuates differentiation into adipocytes. These results implicate epoxides as important disruptors of adipose tissue functions required for metabolic health.

**Decision letter after peer review:**

Thank you for submitting your article "EPHX1 mutations cause a lipoatrophic diabetes syndrome due to impaired epoxide hydrolysis and increased cellular senescence" for consideration by *eLife*. Your article has been reviewed by 3 peer reviewers, one of whom is a member of our Board of Reviewing Editors, and the evaluation has been overseen by David James as the Senior Editor. The following individual involved in review of your submission has agreed to reveal their identity: Robert Semple (Reviewer #3).

Essential revisions:

This manuscript presents novel and important information on a new form of human lipodystrophy related to mutations in EPHX1, and proposes a likely mechanistic hypothesis on how this occurs based on modest support from in vitro studies. The reviewers are in general agreement that this study is meritorious and advances the field. A revised manuscript is anticipated. We have left virtually the entire texts of the reviewers for your considerations in the revision, but they are related to only 3 issues that should need to be considered or addressed.

1) As outlined in Reviewer 3's first 5 bullet points and in Reviewer 2's comments for revision, more information would be needed on the clinical phenotype and genetics of the patients analyzed. Please see the details of these requests in their reviews.

2) As outlined by Reviewer 1, might epoxide content be analyzed in human adipocytes in order to use a more relevant in vitro model and additional mechanistic information that may more strongly support the overarching hypothesis?

3) Some technical issues in data presentation, as outlines in Reviewer 3's bullet points 6 and 7

4) Off target effects of the CRISPR-based deletions would be required to assess the possibility (which is low) that the effects you observed in cultured adipocytes are due to toxic non-specific gene modificaitons by the CRISPR procedure. (Reviewer 2). This is standard practice in the field.

*Reviewer #1:*

A major strength of this study is the identification of novel mutations in EPHX1 that are linked to lipodystrophy in humans. Mechanistic data are also nicely provided, suggesting dysfunctional EPHX1 in adipocytes is detrimental to differentiation and the viability of those adipocytes that do differentiate. The data suggest the mechanism of toxicity due to EPHX1 mutation is the inability of such adipocytes to remove epoxides, and the m vitro data are supportive of this hypothesis.

Further evidence in support of the major hypothesis would be the possible identification of epoxides that may accumulate in adipocytes with mutated EPHX1. Also it would be of interest to show effects of EPHX1 disruption in human adipocytes rather than relying on the mouse 3T3-L1 cells.

Nonetheless, the study is novel, mechanistically insightful and advances the field significantly.

1. Further evidence in support of the major hypothesis would be the possible identification of epoxides that may accumulate in adipocytes with mutated EPHX1. Have the authors performed lipidomic analysis of the adipocytes that are deficient in EPHX1? Could epoxide content be measured?

2. It would be of interest to show effects of EPHX1 disruption in human adipocytes rather than relying on the mouse 3T3-L1 cells. This could be done with human preadipocytes using Cas9/sgRNA RNPs, which yield high efficiency knockdowns without the need for selection of cells.

*Reviewer #2:*

1. The key starting point for this work is the human genetics, so is it convincing. In general, I think it is highly suggestive but clearly a few more cases with EPHX1 mutations would be helpful. In the absence of this I suggest that some additional information be shared. A standard trio analysis such as that performed herein should include consideration of any homozygous variants, compound heterozygous variants and all de novo variants. This should be provided for both cases so that readers can review these data too.

2. Regarding the clinical phenotyping – this is reasonable but was more limited in the second case – there are some clinical points which ought to be addressed. Specifically, a full description of the patient's fat distribution should be included and ideally fat mass (DXA), leptin and adiponectin ought to be measured, though I appreciate this may not be feasible for some reason. Either way it should be overtly clarified.

– The image in figure 1D is not clear enough to show lipodystrophy – is it possible to improve on this or show images of the fat distribution – even if only from the DXA scan.

3. The bioinformatic structural predictions of the impact of the mutations and the impact on enzyme activity look convincing though I am not familiar with the assay.

4. The functional studies described in figure 3 are suboptimal as I worry that without the arrows in 3A, it might not be easy for a reader to see the differences. I also wonder if an 'aggregate' would really run as a clean single band as shown in the gels in 3B.

It might help to see more zoomed in views of the cells and for the authors to clarify whether or not 'aggregates' were also apparent in cells expressing the other mutations. If they were, then some form of quantification should be included.

5. CRISPR/Cas9 mediated gene deletion is widely felt to be challenging in 3T3L1 adipocytes as they are typically aneuploid. Then whilst the data clearly shows impaired differentiation, the underlying mechanism is unclear. I don't think this is formally required at this stage as this can be challenging to pin down conclusively but can the authors be certain that the impact is not due to an off target effect of the CRISPR strategy? Did they sequence the gene etc to confirm the expected impact?

The claim that this is a novel phenotype caused by EPHX1 mutations seems very likely to be true and the functional work is probably adequate for an initial publication. Ultimately more similar cases will be needed to validate the claims. I have highlighted some technical issues, the most important one being inclusion of more sequencing detail.

*Reviewer #3:*

In this manuscript Gautheron and colleagues describe two people with heavily overlapping clinical problems including insulin resistance, reduced adipose tissue (lipodystrophy) and features of "lipotoxicity" (e.g. fatty liver, high serum triglyceride concentration). Other shared syndromic features include an abnormal facial appearance, and sensorineural deafness. Both probands were found to harbour de novo missense mutations in EPHX1 (i.e. the mutation is absent from the parents). EPHX1 encodes a microsomal epoxide hydrolase known to hydrolyse a range of xenobiotic and endogenous epoxides, which are a source of redox stress and cellular damage. Overexpression studies in HEK293 cells with biochemical assay of hydrolysis of a labelled substrate convincingly demonstrate nearly complete loss of function of the missense mutations identified in the probands, in keeping with structural modelling suggesting important roles in stabilising the active site of the enzyme. Overexpressed mutant but not wild type enzyme appeared to aggregate in the endoplasmic reticulum, giving rise to speculation that they might exert a dominant negative effect over co-expressed wild type.

In further studies, Crispr/CAS9 editing was used to knock out Ephx1 in 3T3-L1 cells, a clonal mouse embryo-fibroblast preadipocyte model. After selection of a polyclonal population of cells with knockdown, it was demonstrated that knockout but not control cells could differentiate into mature adipocytes, with reduced insulin responsiveness. Finally, both knockout 3T3-L1 cells and primary dermal fibroblasts were shown to have some increased markers of replicative senescence, including reduced proliferation, increased SA β-gal staining, increased reactive oxygen species, increased phosphorylation of p53, and increased protein expression of p21 and p16.

Overall, this study will be of interest in particular to genetecists, metabolic physicians, adipose biologists, and to those studying ageing. Although the argument is not developed in detail, loss of EPHX1 could confer lipodystrophy and insulin resistance in at least 2 ways. First, by allowing accumulation of toxic ROS, it may increase DNA damage and cellular ageing in adipose precursor or mesenchymal stem cells. Disorders such as Werner syndrome already suggest that this lineage is particularly vulnerable to DNA damage. It is also possible that EPHX1 deficiency reduces generation of an important endogenous ligand of PPARG, the master adipogenic transcription factor, though this is more difficult to prove.

The evidence in this study that EPHX1 mutations cause the syndrome described is moderately convincing but could be enhanced, and the variants found clearly abrogate enzyme function. The evidence for a dominant mechanism mechanism of action is circumstancial only, while some aspects of the phenotypic description seem an overreach based on the data presented. I would add the following specific observations:

Clinical phenotype:

1. There is quite a lot of discussion of "steroidogenesis abnormalities" but there is no clear evidence offered of anything more than is common seen in severe insulin resistance (IR), when ovarian androgen production may be extremely elevated. Please could the authors place their observations in the context of other forms of severe IR? Is there any evidence that the steroid profile is different?

2. EPHX1 expression is very high in the adrenal glands, and it is commented that these are normal in volume and morphology on MRI. But what about their function? Was synacthen testing or other profiling of adrenal steroids undertaken?

3. It is implied that leptin may be a specific treatment for this disorder. In fact the response to leptin is entirely as expected from this degree of lipodystrophy and baseline serum leptin. Parenthetically, it is claimed that the serum leptin concentration (4mcg/L) is consistent with generalised lipodystrophy, whereas to me this looks much more in keeping with known forms of partial lipodystrophy.

Evidence for causation of syndrome by EPHX1 mutations:

4. Raw exome data are not made available as part of this study. This is commonly done in human genetic studies, and I assume that when the authors say it cannot be done here, they mean that suitable consent is not in place. This is reasonable but should be explicit. Without such raw data it would be helpful to readers to have more genetic information. The headline finding of a de novo EPHX1 mutation is given, but were there any other de novo, compound heterozygous or homozygous, rare, likely loss of function mutations in the first proband studied?

5. In the GnomAD repository (up to around 250,000 alleles), there is no evidence of selection against EPHX1 loss of function variants in the general population. I counted 66 loss-of-function alleles (nonsense/frameshift/essential splice site) and a large number of missense variants, including at least one affecting a residue in the catalytic triad. It is implausible, to me, that this syndrome occurs in all these people, and this suggests that heterozygous EPHX1 loss-of-function mutations are likely only rarely associated with the syndrome described, either because of low penetrance, or because simple loss of function of one allele is insufficient. This gives circumstancial support for the notion that only certain mutations that produce dominant negative activity may be required for disease. This does not undermine the case made in the manuscript, but may be worthy of comment.

Evidence for mechanism linking EPHX1 mutations to lipodystrophy:

6. Biochemical studies in overexpressing HEK293 cells show convincingly that the EPHX1 mutations nearly abolish enzyme activity

7. 3T3-L1 knockout studies in mixed populations appear to have been well executed, and the finding that adipogenesis is impaired with increased senescence markers seems sound. Nevertheless 3T3-L1 cells are derived from mouse embryonic fibroblasts, and it is notable that Ephx1 knockout mouse were reported to be phenotypically normal (J. Biol. Chem. 274: 23963-23968). This is worth discussing.

8. Although insulin signalling appears impaired, is this simply a consequence of impaired differentiation, when expression of the insulin receptor normally increases? What about insulin signalling in preadipocytes? Any reduction could also be an indirect consequence of senescence. Although the patients have systemic insulin resistance, this does NOT necessarily need to correspond to a cell autonomous defect in insulin action. Adipose dysfunction would suffice to explain the clinical derangement.

9. The evidence for a dominant negative mechanism of action of the mutant enzymes is circumstancial only, though very plausible. The aggregates seen in HEK293s may be relevant, but this is not clear in such overexpression studies. Nevertheless it would appear straightforward to conduct some further studies in this model, looking at the ability of co-expressed mutant to reduce activity of co-expressed wild type enzyme, as long as care is taken to include suitable controls.

10. One of the challenges in studying primary dermal fibroblasts is that "passage 1" is often timed from establishment of outgrown cells in the lab, which in turn may take weeks from the time of tissue biospy. Could the authors clarify that the cells and controls used had roughly been through the same length of time and/or doubling times in culture?

11. Was there evidence of increased DNA damage in affected cells?

12. Speculation that loss of Ephx1 may alter generation of endogenous agonists (and maybe) antagonists of PPARG is interesting but untested.

13. More detail on birthweights and growth parameters would be useful for both probands. Was the microcephaly sustained in the first patient? What about the second patient?

14. Please address the other clinical questions above.

15. Please list other de novo, compound heterozygous or homozygous, rare, likely loss of function mutations in the first proband and add a statement in text that all parents and any siblings were clinically unaffected.

16. I assume that EPHX1 mutations were also sought in other lipodystrophy cohorts but weren't found. Can any information be offered about this?

17. Through GeneMatcher were any other EPHX1 mutations reported linked to different phenotypes?

18. HEK293 studies were undertaken as duplicates of n=2. Showing data as mean +/- SEM is inappropriate. Better just show points as a scatter plot (see below).

19. "dynamite plunger" plots when numbers are small can hide important data heterogeneity and are now widely disfavoured. Please could all data points be shown superimposed as a scatter plot on these graphs.

20. Language: generally good but with small grammatical lapses (e.g. in abstract). Also, even for this clinically qualified reader some of the technical jargon is obscure and not widely used in anglophone medicine. An "ogival palate" would usually be a "high-arched palate" and "spaniomenorrhoea" is better called "oligomenorrhoea". There are other examples of whether more routine medical jargon could be made more accessible for non-clinical readers, too.

---

## [Author Response]

Essential revisions:This manuscript presents novel and important information on a new form of human lipodystrophy related to mutations in EPHX1, and proposes a likely mechanistic hypothesis on how this occurs based on modest support from in vitro studies. The reviewers are in general agreement that this study is meritorious and advances the field. A revised manuscript is anticipated. We have left virtually the entire texts of the reviewers for your considerations in the revision, but they are related to only 3 issues that should need to be considered or addressed.1) As outlined in Reviewer 3's first 5 bullet points and in Reviewer 2's comments for revision, more information would be needed on the clinical phenotype and genetics of the patients analyzed. Please see the details of these requests in their reviews.

Genetic issue:

The Reviewers 2 and 3 requested the presentation of all alternative genetic hypotheses that could be considered to explain the disease in patients presented in this study. To address this important issue, we have added two Supplementary Files listing the characteristics of all other rare variants identified in the two families investigated herein, according to the corresponding mode of inheritance. For each variant, we indicated the items arguing against its involvement in the disease phenotype. For both patients, we did not find any likely alternative molecular etiology. This is now stated in the Results section of the manuscript (Page 5) as follows: “We did not identify any alternative molecular etiology compatible with the disease phenotype in either of the two patients. A detailed list of the other rare de novo, compound heterozygous, and homozygous variants, as well as the reasons for their exclusion is provided in Supplementary File 1 and 2.”

Clinical issue:

The reviewers also requested more clinical information. We have collected as much information as possible, while respecting the patients' willingness to perform additional investigations. In particular, patient 2 requested no publication of her photos.

Please find below the clinical data that have been added into the revised manuscript:

– Birthweights and growth parameters

– For patient 1:

Page 6: “This patient (woman) was born at term after a normal pregnancy without intra-uterine growth retardation. The anthropometric parameters at birth were normal with a height of 49 cm, and a weight of 2.8 kg. She was first referred for dysmorphic features including microcephaly with an occipito-frontal circumference (OFC) of 33 cm at birth (-1.5 SD), which remained present in adulthood with an OFC of 51 cm (-2.5 SD) at the age of 18 years.”

– For patient 2:

Page 7: “This patient (woman) was born at term, after a normal pregnancy, with a height of 50 cm and a weight of 3.2 kg.”

– Lipoatrophic phenotype

A dual-energy x-ray absorptiometry (DXA)-scan was performed in patients 1 and 2 providing whole body and segmental measures of fat percentage and further confirming the lipoatrophic phenotype. This is now detailed in the Results section of the revised manuscript:

– For patient 1 (Page 6): “This lipoatrophic phenotype was further confirmed by dual X-ray absorptiometry (DXA) with a total fat mass of 15.8%, whereas the mean normal age-matched value is 31.4 ± 8.5% (23), corresponding to a Z-score of -2.8. The study of segmental body composition revealed that the loss of adipose tissue was evenly distributed throughout the body (Figure 1—figure supplement 2).”

– For patient 2 (Page 7): “Lipoatrophy was first noted in the face and the lipoatrophic phenotype was further confirmed by DXA with a total fat mass of 12.4%, a value within the first percentile as compared to age-matched normal individuals. The study of segmental body composition revealed that the loss of adipose tissue affected the whole body and was more pronounced in upper and lower limbs (Figure 1—figure supplement 3).”

Two figure supplements to figure 1 have also been added presenting DXA results, detailed body composition, and adipose indices.

Values for additional biological parameters have been obtained and added in the revised version of the manuscript (Page 7) as follows: “Measurement of serum levels of leptin (3 ng/mL) and adiponectin (0.3 mg/L) further confirmed the lipoatrophic and insulin-resistant phenotype.” These values have also been added in Table 1.

– Adrenal steroid profiling

Results of adrenal steroid profiling, performed several times for patient 1, have been added in the revised version of the manuscript (Page 7): “Adrenal steroid profiling revealed normal levels of cortisone, cortisol, 21-desoxycortisol, 11-desoxycortisol, aldosterone, corticosterone, 21-desoxycorticosterone, 11-desoxycorticosterone, and ACTH.”

2) As outlined by Reviewer 1, might epoxide content be analyzed in human adipocytes in order to use a more relevant in vitro model and additional mechanistic information that may more strongly support the overarching hypothesis?

Measurement of epoxides is an important point and several complementary approaches were considered to address this issue.

– None of the two patients wished to undergo an adipose tissue biopsy, which could have been used to study adipocyte stem cells (ASCs). Moreover, given that the fat mass was very low in patient 1, the clinicians in charge of the patient considered that the sampling would be very painful and would certainly not make possible to collect a sufficient quantity of tissue.

– We have thus measured the levels of epoxy fatty acids and corresponding diols in the plasma of patient 1. These experiments have been performed by the team of Christophe Morisseau and Bruce Hammock, co-authors of the paper and internationally recognized experts in the field. These data have been added in the revised version of the manuscript (Page 9) together with an additional Figure 2—figure supplement 1 as follows: “To evaluate the impact of the loss of enzyme activity in vivo, we measured by liquid chromatography (LC) coupled with tandem mass spectrometry (MS/MS) circulating levels of a panel of epoxy-fatty acids (EpFAs) and corresponding diols in plasma samples of patient 1. […] Since oxylipin profiling is an emerging field, whose biological interpretation remains difficult, further experiments will be required to confirm this observation in additional patients and/or different cellular models.”

– Since we could not measure EpFAs in cell lines due to the low levels of fatty acids and the requirement of hundred million cells, we have measured the hydrolysis of cis-stilbene oxide (c-SO), a well-characterized substrate of EPHX1, in lysates of *Ephx1* KO 3T3-L1 cells, as compared to WT and control (scramble RNA guide) 3T3-L1 cells. As shown in the new Figure 4—figure supplement 3, KO cells exhibit a strong and significant reduction of c-SO hydrolysis when compared with WT and control 3T3-L1 cells (~60% reduction). This information, which strengthens the relevance of the 3T3-L1 cellular model, has been added in the revised version of the manuscript (Page 10) as follows: “Consistently, hydrolysis of [3H]-cSO was evaluated in cell lysates and revealed a significant loss of enzyme activity in 3T3-L1 KO cells, as compared to control cells (Figure 4—figure supplement 3)”.

3) Some technical issues in data presentation, as outlines in Reviewer 3's bullet points 6 and 7.

We agree with Reviewer #3 that it was inappropriate to show Figure 2D with only two independent biological experiments. We have now repeated this experiment a third time and we could confirm that the two variants identified in patients, p.Thr333Pro and p.Gly430Arg, led to an absence of the enzyme activity as compared to the WT protein and the three other isoforms carrying variants from the general population. We have enriched Figure 2D with these additional data, but we did not replace all graphs by scatter plots since all the other experiments were conducted with at least three independent assays.

4) Off target effects of the CRISPR-based deletions would be required to assess the possibility ( which is low) that the effects you observed in cultured adipocytes are due to toxic non-specific gene modificaitons by the CRISPR procedure. (Reviewer 2). This is standard practice in the field.

Off-target effects have been reported for various CRISPR effectors, including Cas9, which was used in our cellular system. As detailed below, we have now designed three different RNA guides (gRNA) and generated two murine and one human *EPHX1* KO cellular models. The three of them led to similar cellular alterations. This is a major point to demonstrate that the effects observed are not due to off-target mutations. In addition, we strictly followed the recommendation of several current genome editing protocols to select gRNA carefully in order to avoid off-target effects and to ensure high cleavage efficiency (Ran FA et al. Nat Protoc 2013 Nov;8(11):2281-308). Please find below, the list of measures undertaken to prevent off-target activities in each cellular model.

For the gRNA targeting *Ephx1* exon 6, used in murine 3T3-L1 cells in the first version of the manuscript:

– The CRISPOR web tool (http://crispor.tefor.net/) is well recognized to predict the risk of off-target sequences by providing a cutting frequency determination (CFD) specificity score ranging from 1 to 100. The higher the number, the lower the risk of off-target effects. It is based on the accurate CFD off-target model from Doench JG et al. (Nat Biotechnol 2016 Feb;34(2):184-196,) which recommends guides with a CFD specificity score > 50. The gRNA targeting exon 6 has a CFD score of 88. First of all, the gRNA used did not match perfectly any other genomic region outside of the *Ephx1* locus. Please find below a list of off-target sequences for this gRNA with up to three mismatches as compared to our gRNA sequence (TCTTAGAGAAGTTCTCCACCTGG). Notably, off-targets are considered if they are flanked by an NGG motif, which corresponds to the PAM sequence allowing the Cas9 to cut DNA.

**Author response table 1. resptable1:** 

Number of mismatches	Potential off-target sequences (mismatches are in red and bold characters)	Locus of the off-target (gene / location)
2	TCTTAGTGAAGTGCTCCACCTAG	*Zfhx3* / intron
3	TTTTAGAGAAGTTGACCACCTGG	*Gdpgp1* / exon
3	TAATAGAGAAGTTCTCGACCTGA	*Trpc7* / intron
3	TACTAGAGAAGTTCTCCAGCTGA	intergenic
3	TCTCAGCCAAGTTCTCCACCAAG	intergenic
3	TCTCAGACATGTTCTCCACCAAG	intergenic
3	TCTTGGAGAAGTTCTTCAACAGG	intergenic
3	TCTTAGATAATTTCTCAACCAGG	intergenic
3	TCTTAGAGAAGTTTACCACTAGG	intergenic
3	TCCTAGAGAATTCCTCCACCTGG	intergenic
3	CCTTGGAGATGTTCTCCACCCAG	*Fign* / intron
3	TCTAGGAGAAGTTCTCCACAAGG	intergenic
3	TCTTGGAGAAGTCCTTCACCTGG	intergenic
3	TGTTACAGAAGTTCTCAACCTGG	intergenic
3	TTTCAGAGAAGTTCTCTACCAGG	intergenic
3	GCTGAGAGAAGTTCTCCACAAGG	*Clnk* / intron

We have clarified this point into the Methods section of the revised manuscript (Page 22) as follows: “We used the web-based tool, CRISPOR () to avoid off-target effects.”

– To avoid off-target activities of Cas9, we privileged a plasmid transient transfection method rather than a viral transduction method to (i) reduce the expression of Cas9 in cells and (ii) prevent its integration into the genome of host cells. Indeed, it was demonstrated that extended expression of Cas9 in cells can lead to accumulation of off-targeting events (Kim S et al. Genome Res 2014 Jun; 24(6):1012-1019). Also, we favored the use of GFP to positively select transfected cells rather than an antibiotic resistance cassette, which would allow the integration of the plasmid into the genome (although the frequency of this event would be very low).

We have updated this point into the Methods section of the revised manuscript (Page 23) as follows: “We favored a plasmid transient transfection method rather than a viral transduction to reduce the expression of Cas9 in cells and prevent its integration into the host cell genome, which may lead to increased off-target activities.”

– To minimize the effect of possible off-target mutations, we analyzed heterogeneous populations issued from the FACS sorting rather than clonal populations (i.e., 10.997 sorted cells for the knock-out cell line). It is unlikely that the same off-target activities would occur in all cells.

This strategy to minimize off-target effects has been clarified into the Methods section of the revised manuscript (Page 23) as follows: “Moreover, to minimize the effect of possible off-target mutations, we analyzed heterogeneous populations issued from the FACS sorting rather than clonal populations.”

– To avoid the delivery of ineffective gRNA in our cells, we tested the gRNA efficacy in vitro. As expected from the CRISPOR designer tool, synthesized gRNA associated with recombinant Cas9 was able to cleave PCR products comprising *Ephx1* exon 6 in vitro.

**Author response image 1. respfig1:** Line 1: untreated EPHX1 PCR fragments; Line 2: EPHX 1 PCR fragments trated with gRNA and recombinant Cas9 enzyme.

Second gRNA targeting *Ephx1* exon 5, used in murine 3T3-L1 cells in the revised version of the manuscript:

– We have designed another gRNA, which targets *Ephx1* exon 5. Its CFD specificity score was 75 and, of note, the potential off-targets were completely different to those related to the initial gRNA.

The results related to this additional gRNA have been added and discussed into the revised manuscript (Page 11) as follows: “To exclude the possibility that undesired off-target mutations were responsible for the effects observed in KO cells, we used another gRNA, which targets *Ephx1* exon 5. […] As revealed by Oil Red O staining, this new KO cellular model had a similar defect in adipocyte differentiation as the first KO cell line used throughout this study (Figure 4—figure supplement 5).”

Third gRNA targeting *Ephx1* exon 5, used in human adipose stem cells (ASCs) in the revised version of the manuscript:

– To comply with Reviewer #1’s suggestion, we have knocked down *EPHX1* in human ASCs using a third gRNA targeting *EPHX1* exon 3 with a CFD score of 77. We could recapitulate the senescence phenotype observed in 3T3-L1 KO cells and in fibroblasts from patient 1. The level of cellular senescence was extremely high preventing the KO cells to be differentiated into adipocytes. These novel data in primary human cells confirm a functional link between EPHX1 and cellular senescence, which may underlie the lipodystrophy phenotype. The results have been added in the Results section of the manuscript (page 12) as follows: “To further demonstrate the relevance of the 3T3-L1 murine model, a lentiviral CRISPR/Cas9-mediated *EPHX1* KO was generated in human adipose stem cells (ASCs) using a custom-designed gRNA targeting the third exon of *EPHX1*. […] Altogether, these data obtained in a murine cell line and validated in a human cellular model strongly argue for a functional link between EPHX1 dysfunction, oxidative stress, and cellular senescence.”

Collectively, the gRNA used in this study appears to be highly specific and it is unlikely that the observed cellular effects are due to unwanted off-target mutations.

Reviewer #1:A major strength of this study is the identification of novel mutations in EPHX1 that are linked to lipodystrophy in humans. Mechanistic data are also nicely provided, suggesting dysfunctional EPHX1 in adipocytes is detrimental to differentiation and the viability of those adipocytes that do differentiate. The data suggest the mechanism of toxicity due to EPHX1 mutation is the inability of such adipocytes to remove epoxides, and the m vitro data are supportive of this hypothesis.Further evidence in support of the major hypothesis would be the possible identification of epoxides that may accumulate in adipocytes with mutated EPHX1. Also it would be of interest to show effects of EPHX1 disruption in human adipocytes rather than relying on the mouse 3T3-L1 cells.Nonetheless, the study is novel, mechanistically insightful and advances the field significantly.

We would like to thank Reviewer #1 for his/her kind words and detailed evaluation of our manuscript. Based on his/her comments, we have added several experiments to show the effect of *EPHX1* mutations in patient 1 and to strengthen the mechanistic aspects of our study.

Please find below our point-by-point response.

1. Further evidence in support of the major hypothesis would be the possible identification of epoxides that may accumulate in adipocytes with mutated EPHX1. Have the authors performed lipidomic analysis of the adipocytes that are deficient in EPHX1? Could epoxide content be measured?

We agree with Reviewer #1 that measurement of epoxides is an important point and several complementary approaches were considered to address this issue. For more details, please also refer to comment 2 of the Editor. To summarize:

– None of the two patients wished to undergo an adipose tissue biopsy. Moreover, given that the fat mass was very low in patient 1, the clinicians in charge of the patient considered that the sampling would be very painful and would certainly not make possible to collect a sufficient quantity of tissue.

– We have measured the levels of several epoxy fatty acids and corresponding diols in the plasma of patient 1. These experiments have been performed by the team of Christophe Morisseau and Bruce Hammock, co-authors of the paper and internationally recognized experts in the field. These data have been added in the revised version of the manuscript together with a Figure 2—figure supplement 1.

– Measurement of EpFAs in cell lines is a very hard task due to the low amount of fatty acids and the need of hundred million cells. This would have been all the more difficult as the *Ephx1* KO 3T3-L1 do not differentiate into adipocytes. However, to strengthen the relevance of the cellular model used, we have measured the hydrolysis of *cis*-stilbene oxide (c-SO), a well-characterized substrate of EPHX1, in lysates of 3T3-L1 cells. As shown in the new Figure 4—figure supplement 3, *Ephx1* KO cells exhibit a strong and significant reduction of c-SO hydrolysis as compared to WT and control (scramble gRNA) 3T3-L1 cells (~60% reduction). This information has been added in the revised version of the manuscript together with a Figure 4—figure supplement 3. Please also refer to comment 2 of the Editor for more details.

2. It would be of interest to show effects of EPHX1 disruption in human adipocytes rather than relying on the mouse 3T3-L1 cells. This could be done with human preadipocytes using Cas9/sgRNA RNPs, which yield high efficiency knockdowns without the need for selection of cells.

As suggested by Reviewer #1, we thought of using adipose stem cells (ASCs), which can be differentiated into adipocytes. The standard differentiation protocol requires the use of an adipogenic induction cocktail containing insulin, glucocorticoids (e.g., dexamethasone), and 1-methyl-3-isobutylxanthine (IBMX) (Lee M-J et al. Methods Enzymol. 2014;538:49-65). In addition, PPARγ agonists, such as rosiglitazone, are often used to improve cell differentiation capacity since human ASCs do not differentiate well in the absence of PPARγ ligands (Ahmadian M et al. Nat Med 2013 May:19(5):10.1038/nm.3159). For this study, we initially excluded the use of ASCs because PPARγ agonists are known to regulate the expression of epoxide hydrolases (De Taye BM et al. Obesity. 2010 Mar;18(3):489-98), notably EPHX2, and functional compensation between EPHX1 and EPHX2 has been shown to occur in vivo (Edin ML et al. J Biol Chem. 2018 Mar2; 293(9):3281-92). Therefore, the use of PPARγ agonists to differentiate ASCs may induce a compensatory phenomenon making the analysis of *EPHX1* KO hazardous. Since the differentiation protocol of 3T3-L1 pre-adipocytes does not require PPARγ agonists, we favored this model, which led to major advances in our understanding of adipogenesis and lipid metabolism over the last decade.

– Nevertheless, to comply with Reviewer #1’s suggestion, we have generated a human ASC model. As stem cells are notoriously difficult to transfect with standard methods/reagents (e.g., electroporation, lipid-based transfection), we favored the use of a lentiviral CRISPR/Cas9 system to knock down *EPHX1*. This additional set of data confirms the observations made in 3T3-L1 cells and fibroblasts derived from patient 1 and the functional link between EPHX1 dysfunction and cellular senescence. Please also refer to comment 4 of the Editor for more details.

The new lentiviral CRISPR/Cas9 system is now described in the Methods section of the revised manuscript (Page 23): “The lentiviral plasmid plentiCRISPRv2 was a gift from Zhang lab (Addgene, MA, USA; plasmid #52961) and contains the puromycin resistance, hSpCas9 and the chimeric guide RNA (gRNAs). […] ASCs were infected with viral particles at a minimal titer of 10^8^ transducing units per mL. 48 h post infection, cells were selected with 5 μg/ml puromycin dihydrochloride (#P9620; Sigma-Aldrich). Surviving cells were propagated and the heterogeneous cell pool was used for experiments.”

The description of ASC isolation and culture has been added in the Methods section of the revised manuscript (Page 22) as follows: “ASCs were isolated from surgical samples of sub-cutaneous abdominal adipose tissue from a control woman of the same sex and age as patient 1 and normal BMI. […] ASCs were maintained in an undifferentiated state in high-glucose (4.5 g/L) DMEM supplemented with 10 % newborn calf serum and PS 1 %.”

The results obtained in these human cells have been added into the revised manuscript (Page 12) as follows: “To further demonstrate the relevance of the 3T3-L1 murine model, a lentiviral CRISPR/Cas9-mediated *EPHX1* KO was generated in human adipose stem cells (ASCs) using a custom-designed gRNA targeting the third exon of *EPHX1*. […] Altogether, these data obtained in a murine cell line and validated in a human cellular model strongly argue for a functional link between EPHX1 dysfunction, oxidative stress, and cellular senescence.”

Reviewer #2:1. The key starting point for this work is the human genetics, so is it convincing. In general, I think it is highly suggestive but clearly a few more cases with EPHX1 mutations would be helpful. In the absence of this I suggest that some additional information be shared. A standard trio analysis such as that performed herein should include consideration of any homozygous variants, compound heterozygous variants and all de novo variants. This should be provided for both cases so that readers can review these data too.

To address this important issue, we have added two Supplementary Files listing the characteristics of all rare de novo, compound heterozygous, and homozygous variants, as well as the reasons accounting for their exclusion. No alternative molecular etiology was identified in either patient. This is discussed in the revised version of the manuscript. For a more detailed response, please see the answer to the first comment of the Editor.

2. Regarding the clinical phenotyping – this is reasonable but was more limited in the second case – there are some clinical points which ought to be addressed. Specifically, a full description of the patient's fat distribution should be included and ideally fat mass (DXA), leptin and adiponectin ought to be measured, though I appreciate this may not be feasible for some reason. Either way it should be overtly clarified.– The image in figure 1D is not clear enough to show lipodystrophy – is it possible to improve on this or show images of the fat distribution – even if only from the DXA scan.

We have collected as much information as possible, while respecting the patients' willingness to perform additional investigations. In particular, patient 2 requested no publication of her photos.

A dual-energy x-ray absorptiometry (DXA)-scan was performed in patients 1 and 2 providing whole body and segmental measures of fat percentage and further confirming the lipoatrophic phenotype. This is now detailed in the Results section of the revised manuscript:

– For patient 1 (Page 6): “This lipoatrophic phenotype was further confirmed by dual X-ray absorptiometry (DXA) with a total fat mass of 15.8%, whereas the mean normal age-matched value is 31.4 ± 8.5% (23), corresponding to a Z-score of -2.8. The study of segmental body composition revealed that the loss of adipose tissue was evenly distributed throughout the body (Figure 1—figure supplement 2).”

– For patient 2 (Page 7): “Lipoatrophy was first noted in the face and the lipoatrophic phenotype was further confirmed by DXA with a total fat mass of 12.4%, a value within the first percentile as compared to age-matched normal individuals. The study of segmental body composition revealed that the loss of adipose tissue affected the whole body and was more pronounced in upper and lower limbs (Figure 1—figure supplement 3).”

Two figure supplements to figure 1 have also been added presenting DXA results, detailed body composition, and adipose indices.

Values for additional biological parameters have been added for patient 2 in the revised version of the manuscript (page 7) as follows: “Measurement of serum levels of leptin (3 ng/mL) and adiponectin (0.3 mg/L) further confirmed the lipoatrophic and insulin-resistant phenotype.” These values have also been added in Table 1.

3. The bioinformatic structural predictions of the impact of the mutations and the impact on enzyme activity look convincing though I am not familiar with the assay.

We thank Reviewer #2 for this comment.

4. The functional studies described in figure 3 are suboptimal as I worry that without the arrows in 3A, it might not be easy for a reader to see the differences. I also wonder if an 'aggregate' would really run as a clean single band as shown in the gels in 3B.It might help to see more zoomed in views of the cells and for the authors to clarify whether or not 'aggregates' were also apparent in cells expressing the other mutations. If they were, then some form of quantification should be included.

We agree with Reviewer #2 that “aggregate” was an imprecise word. We have clarified this within the revised manuscript by changing “aggregates” by “higher-order complexes”, a term better reflecting the observations made by Western blot analysis.

5. CRISPR/Cas9 mediated gene deletion is widely felt to be challenging in 3T3L1 adipocytes as they are typically aneuploid. Then whilst the data clearly shows impaired differentiation, the underlying mechanism is unclear. I don't think this is formally required at this stage as this can be challenging to pin down conclusively but can the authors be certain that the impact is not due to an off target effect of the CRISPR strategy? Did they sequence the gene etc to confirm the expected impact?

We acknowledge that this is an important point and we have now included a new set of experiments to strengthen the validity of our cellular model and to ensure that the effects were independent of off-target mutations. Please refer to point 4 of the Editor for a more detailed response. To summarize:

– We used the powerful and well-recognized tool “http://crispor.tefor.net” to predict and avoid off-targets.

– We favored a transient transfection method and GFP cell sorting to avoid prolonged Cas9 expression, which may enhance off-target effects.

– We analyzed heterogeneous populations issued from the FACS sorting rather than sub-clonal populations to minimize effect of possible deleterious off-target mutations. It is unlikely that a main off-target mutation occurred in all cells.

– We tested the gRNA efficacy in vitro to avoid delivery of ineffective gRNA in 3T3-L1 cells.

– We could confirm a strong and significant reduction of the hydrolysis of *cis*-stilbene oxide (c-SO) in lysates of *EphX1* KO 3T3-L1 cells. This confirms the Western blot analysis and the knock-down efficiency.

– We conducted KO experiments using a different gRNA targeting *Ephx1* exon 5 in 3T3-L1 cells. We could confirm a strong reduction of adipocyte differentiation with this new KO cell line, as seen with the initial gRNA. Since the sequence of this gRNA is completely different from that of the initial one, it argues against off-target effects.

– We generated an additional *EPHX1* KO model in human adipose stem cells (ASCs) using another gRNA targeting the third exon of human EPHX1. A near complete loss of EPHX1 expression was observed by Western blot analysis (Figure 5F). This led to a reduction of the proliferative capacity of the cells and to a major increase of cellular senescence (Figure 5G and 5H) and recapitulated the effects seen in 3T3-L1 cells.

Reviewer #3:In this manuscript Gautheron and colleagues describe two people with heavily overlapping clinical problems including insulin resistance, reduced adipose tissue (lipodystrophy) and features of "lipotoxicity" (e.g. fatty liver, high serum triglyceride concentration). Other shared syndromic features include an abnormal facial appearance, and sensorineural deafness. Both probands were found to harbour de novo missense mutations in EPHX1 (i.e. the mutation is absent from the parents). EPHX1 encodes a microsomal epoxide hydrolase known to hydrolyse a range of xenobiotic and endogenous epoxides, which are a source of redox stress and cellular damage. Overexpression studies in HEK293 cells with biochemical assay of hydrolysis of a labelled substrate convincingly demonstrate nearly complete loss of function of the missense mutations identified in the probands, in keeping with structural modelling suggesting important roles in stabilising the active site of the enzyme. Overexpressed mutant but not wild type enzyme appeared to aggregate in the endoplasmic reticulum, giving rise to speculation that they might exert a dominant negative effect over co-expressed wild type.In further studies, Crispr/CAS9 editing was used to knock out Ephx1 in 3T3-L1 cells, a clonal mouse embryo-fibroblast preadipocyte model. After selection of a polyclonal population of cells with knockdown, it was demonstrated that knockout but not control cells could differentiate into mature adipocytes, with reduced insulin responsiveness. Finally, both knockout 3T3-L1 cells and primary dermal fibroblasts were shown to have some increased markers of replicative senescence, including reduced proliferation, increased SA β-gal staining, increased reactive oxygen species, increased phosphorylation of p53, and increased protein expression of p21 and p16.Overall, this study will be of interest in particular to genetecists, metabolic physicians, adipose biologists, and to those studying ageing. Although the argument is not developed in detail, loss of EPHX1 could confer lipodystrophy and insulin resistance in at least 2 ways. First, by allowing accumulation of toxic ROS, it may increase DNA damage and cellular ageing in adipose precursor or mesenchymal stem cells. Disorders such as Werner syndrome already suggest that this lineage is particularly vulnerable to DNA damage. It is also possible that EPHX1 deficiency reduces generation of an important endogenous ligand of PPARG, the master adipogenic transcription factor, though this is more difficult to prove.The evidence in this study that EPHX1 mutations cause the syndrome described is moderately convincing but could be enhanced, and the variants found clearly abrogate enzyme function. The evidence for a dominant mechanism mechanism of action is circumstantial only, while some aspects of the phenotypic description seem an overreach based on the data presented. I would add the following specific observations:

We thank the Reviewer #3, Prof. Robert Semple, for his detailed and positive evaluation of our manuscript. Based on his comments, we have added a number of additional information and experimental data. We have also thoroughly revised the manuscript in order to gain in clarity and to better discuss our findings in light of the existing literature.

Clinical phenotype:1. There is quite a lot of discussion of "steroidogenesis abnormalities" but there is no clear evidence offered of anything more than is common seen in severe insulin resistance (IR), when ovarian androgen production may be extremely elevated. Please could the authors place their observations in the context of other forms of severe IR? Is there any evidence that the steroid profile is different?

We thank the Reviewer for this interesting question. In searching for specific information to address this issue, we found a very recent publication by the Reviewer 3’s team providing key elements (Huang-Doran I et al. J Clin Endocrinol Metab. 2021 Apr 26). In particular, total testosterone (TT) levels were measured in 173 women with lipodystrophy with a median level of 24.0 ng/dL (interquartile range: 20 – 59 ng/dL), i.e. median: 0.83 nmol/L (interquartile range: 0.69 – 2.04 nmol/L), whereas the TT levels in patient 1 were 16.9 nmol/L. In the revised version of the manuscript (Page 14), we now discuss the potential contribution of insulin resistance in hyperandrogenism signs, in relation to TT levels as follows: “Regarding hormonal pathophysiology, patient 1 developed amenorrhea associated with steroidogenesis alterations. […] In this regard, a potential role of EPHX1 in reproductive physiology was suggested previously.”

2. EPHX1 expression is very high in the adrenal glands, and it is commented that these are normal in volume and morphology on MRI. But what about their function? Was synacthen testing or other profiling of adrenal steroids undertaken?

Results of adrenal steroid profiling, performed several times for patient 1, have been added in the revised version of the manuscript (Page 7) as follows: “Adrenal steroid profiling revealed normal levels of cortisone, cortisol, 21-desoxycortisol, 11-desoxycortisol, aldosterone, corticosterone, 21-desoxycorticosterone, 11-desoxycorticosterone, and ACTH.”

3. It is implied that leptin may be a specific treatment for this disorder. In fact the response to leptin is entirely as expected from this degree of lipodystrophy and baseline serum leptin. Parenthetically, it is claimed that the serum leptin concentration (4mcg/L) is consistent with generalised lipodystrophy, whereas to me this looks much more in keeping with known forms of partial lipodystrophy.

– Metreleptin treatment

We agree that our message on the efficacy of metreleptin treatment was maybe too enthusiastic and could have been misleading to a non-expert reader. We have clarified this point in the Result (Page 13) and Discussion (Page 17) sections as follows: “Treatment with metreleptin, a recombinant form of leptin used in the treatment of lipoatrophic syndromes, was initiated…” and “Metreleptin was shown to reduce hyperphagia leading to weight loss, to improve insulin sensitivity and secretion, to reduce hypertriglyceridemia, hyperglycemia, and fatty liver disease in *many* patients with lipoatrophic diabetes (84,85). All these beneficial effects were rapidly observed in patient 1…”

– Leptin values

Using Reviewer 3’s advice, we have corrected the sentence related to leptin values (Page 6) as follows: “The serum leptin levels, which are strongly correlated with total body fat mass, were very low in patient 1 (4 ng/mL) and similar to those usually reported in *partial* lipodystrophy (24), further confirming the lipoatrophic phenotype.”

Evidence for causation of syndrome by EPHX1 mutations:4. Raw exome data are not made available as part of this study. This is commonly done in human genetic studies, and I assume that when the authors say it cannot be done here, they mean that suitable consent is not in place. This is reasonable but should be explicit. Without such raw data it would be helpful to readers to have more genetic information. The headline finding of a de novo EPHX1 mutation is given, but were there any other de novo, compound heterozygous or homozygous, rare, likely loss of function mutations in the first proband studied?

To address this important issue, we have provided two Supplementary Files listing the characteristics of all rare de novo, compound heterozygous, and homozygous variants, as well as the reasons for their exclusion. No alternative molecular etiology was identified in either patient. This is discussed in the revised version of the manuscript. For a more detailed response, please see the answer to the first comment of the Editor.

5. In the GnomAD repository (up to around 250,000 alleles), there is no evidence of selection against EPHX1 loss of function variants in the general population. I counted 66 loss-of-function alleles (nonsense/frameshift/essential splice site) and a large number of missense variants, including at least one affecting a residue in the catalytic triad. It is implausible, to me, that this syndrome occurs in all these people, and this suggests that heterozygous EPHX1 loss-of-function mutations are likely only rarely associated with the syndrome described, either because of low penetrance, or because simple loss of function of one allele is insufficient. This gives circumstancial support for the notion that only certain mutations that produce dominant negative activity may be required for disease. This does not undermine the case made in the manuscript, but may be worthy of comment.

The reviewer is correct with this comment, and we have better discussed this issue in the revised version of the manuscript (Pages 16-17) as follows: “The gnomAD database, which collects variants from the general population, reports several dozen predicted loss-of-function variants in *EPHX1*, including nonsense, frameshift and canonical splice site variants. […] Additional studies will be required to better understand EPHX1 activity when embedded in the microsomal ER membranes in endogenous conditions.”

Evidence for mechanism linking EPHX1 mutations to lipodystrophy:6. Biochemical studies in overexpressing HEK293 cells show convincingly that the EPHX1 mutations nearly abolish enzyme activity.

We thank Reviewer #3 for this positive comment.

7. 3T3-L1 knockout studies in mixed populations appear to have been well executed, and the finding that adipogenesis is impaired with increased senescence markers seems sound. Nevertheless 3T3-L1 cells are derived from mouse embryonic fibroblasts, and it is notable that Ephx1 knockout mouse were reported to be phenotypically normal (J. Biol. Chem. 274: 23963-23968). This is worth discussing.

We agree with Reviewer #3 that the fact that *Ephx1* knock-out mice are phenotypically normal is worth being discussed. This information has been discussed into the revised manuscript (Page 17) as follows: “Regarding available animal models, *Ephx1* knock-out mice have already been generated (80). […] Knock-in mice would be required to better investigate such pathophysiological mechanisms for *EPHX1* variants, and additional metabolic stress is sometimes needed to uncover more aspects of the human phenotype (83).”

8. Although insulin signalling appears impaired, is this simply a consequence of impaired differentiation, when expression of the insulin receptor normally increases? What about insulin signalling in preadipocytes? Any reduction could also be an indirect consequence of senescence. Although the patients have systemic insulin resistance, this does NOT necessarily need to correspond to a cell autonomous defect in insulin action. Adipose dysfunction would suffice to explain the clinical derangement.

We acknowledge that impaired insulin signaling may be a simple consequence of impaired differentiation, we thus investigated insulin signaling in pre-adipocytes. These data have been included, together with a Figure 4—figure supplement 4, into the revised manuscript (Page 11) as follows: “In contrast, the *Ephx1* KO cells were resistant to insulin, both in pre-adipocytes and differentiated cells, as shown by the lack or strong decrease in the phosphorylation of these intermediates upon insulin stimulation (Figure 4F and Figure 4—figure supplement 4)”.

9. The evidence for a dominant negative mechanism of action of the mutant enzymes is circumstantial only, though very plausible. The aggregates seen in HEK293s may be relevant, but this is not clear in such overexpression studies. Nevertheless it would appear straightforward to conduct some further studies in this model, looking at the ability of co-expressed mutant to reduce activity of co-expressed wild type enzyme, as long as care is taken to include suitable controls.

As suggested by Reviewer #3, we have investigated the ability of co-expressed mutant EPHX1 isoforms to reduce the activity of the wild-type (WT) protein. We first transfected increasing amounts of plasmids encoding the WT and mutated forms of EPHX1 in HEK293 cells, and examined the ability of cell lysates to hydrolyze [^3^H]-cSO substrate, as described in the first version of the manuscript. The total amount of transfected plasmid was kept constant using an empty vector (2 μg). We could confirm that the two variants identified in patients completely abolish the enzyme activity, whereas the WT isoform catalyzes c-SO hydrolysis in a dose-dependent manner. However, we could not observe a dominant negative effect when we co-expressed WT and mutated isoforms. For example, the hydrolysis activity of lysates co-expressing WT EPHX1 (1μg) and one mutated isoform (1μg) was roughly the same as that of lysates transfected only with WT EPHX1 (1μg).

**Author response image 2. respfig2:** 

Of note, this cellular assay consisted of the hydrolysis of a radioactive synthetic substrate and was performed on protein lysates in overexpression studies. Thus, we cannot exclude that the mutants will exert a dominant negative effect on WT EPHX1 in vivo, since the dynamics of protein interaction in the endoplasmic reticulum under endogenous conditions might be different. In this regard, it has been proposed that EPHX1 might aggregate into oligomers (Zhou J, et al., Structure. 2000 Feb 15;8(2):111-22). Technical advances are needed to further investigate this dominant negative effect. For example, good antibodies and relevant assays to visualize hydrolysis activity in cells, such as Bioluminescence Resonance Energy Transfer, need to be developed.The possibility that mutants may exert a dominant negative effect on the WT protein in vivo is better addressed in the revised manuscript (Pages 16-17) and a Figure 2—figure supplement 2 has been added: “The gnomAD database, which collects variants from the general population, reports several dozen predicted loss-of-function variants in *EPHX1*, including nonsense, frameshift and canonical splice site variants. […] Additional studies will be required to better understand EPHX1 activity when embedded in the microsomal ER membranes in endogenous conditions.”

10. One of the challenges in studying primary dermal fibroblasts is that "passage 1" is often timed from establishment of outgrown cells in the lab, which in turn may take weeks from the time of tissue biospy. Could the authors clarify that the cells and controls used had roughly been through the same length of time and/or doubling times in culture?

We confirm that dermal fibroblasts from patients and controls spent roughly the same length of time in culture for the establishment of each cell culture. They were cryopreserved at passage 2. For the experiment presented in the manuscript, fibroblasts from controls were at passage 9 while those from patient 1 at passage 4. This is described in the Methods section (Page 22).

11. Was there evidence of increased DNA damage in affected cells?

Compelling evidence demonstrate that DNA damage is a common mediator for both replicative senescence, which is triggered by telomere shortening, and premature cellular senescence induced by various stressors such as oncogenic and oxidative stress (Chen J-H et al. Nucleic Acids Res. 2007 Dec;35(22):7417-28). Reviewer #3 is certainly right that DNA damage may occur in both fibroblasts of patient 1 and 3T3-L1 knock-out cells. However, we believe that DNA damage investigation may distract the reader from our main messages, and that these experiments should be part of a follow-up study.

12. Speculation that loss of Ephx1 may alter generation of endogenous agonists (and maybe) antagonists of PPARG is interesting but untested.

As stressed by Reviewer #3, the link between the lack of EPHX1 activity, the adipogenesis defect and the deregulation of PPARγ signaling is an interesting perspective to this study. We have better discussed this possibility (Page 15) as follows: “What is the cellular link between the loss of EPHX1 activity and adipogenesis defect? EPHX1 substrates might play a key role since oxylipins, which are EPHXA substrates, target peroxisome proliferator-activated receptors (PPARs) to modify adipocyte formation and function (67). […] Additional experiments will be required to precisely define the link between the loss of EPHX1 and adipogenesis alteration, which might involve the deregulation of endogenous PPARγ agonists or antagonists.”

13. More detail on birthweights and growth parameters would be useful for both probands. Was the microcephaly sustained in the first patient? What about the second patient?

We have added information in the Results section of the revised manuscript (Pages 5-6) as follows:

For patient 1: “This patient (woman) was born at term after a normal pregnancy without intra-uterine growth retardation. […] She was first referred for dysmorphic features including microcephaly with an occipitofrontal-frontal circumference (OFC) of 33 cm at birth (-1.5 SD), which remained present in adulthood with an OFC of 51 cm (-2.5 SD) at the age of 18 years.”

For patient 2:

“This patient (woman) was born at term, after a normal pregnancy, with a height of 50 cm and a weight of 3.2 kg.”

14. Please address the other clinical questions above.

This has been done to the best of our ability, while respecting the patients' wishes in terms of clinical investigations and publication of photos.

15. Please list other de novo, compound heterozygous or homozygous, rare, likely loss of function mutations in the first proband and add a statement in text that all parents and any siblings were clinically unaffected.

To address this important issue, we have provided two Supplementary Files listing the characteristics of all rare de novo, compound heterozygous, and homozygous variants, as well as the reasons accounting for their exclusion. No alternative molecular etiology was identified in either patient. This is discussed in the revised version of the manuscript. For a more detailed response, please see the answer to the first comment of the Editor. Moreover, two sentences have been added (page 7) to explain that parents of patients 1 and 2 are clinically unaffected: “Her parents were clinically unaffected.” / “The parents of patient 2 were clinically unaffected.”

16. I assume that EPHX1 mutations were also sought in other lipodystrophy cohorts but weren't found. Can any information be offered about this?

We looked in our exome results for patients with a genetically-unexplained lipodystrophic syndrome. None of them had a phenotype matching that of the two patients presented in the manuscript, and we identified no rare variants in *EPHX1*. To have a more systematic approach in order to determine the frequency of lipoatrophic diabetes due to a molecular defect in *EPHX1*, this gene is now included in our gene panel and will be analyzed in all patients referred to our laboratory for genetic diagnosis.

17. Through GeneMatcher were any other EPHX1 mutations reported linked to different phenotypes?

There was a single match on GeneMatcher between Isabelle Jéru and Wendy Chung, co-authors of the paper and in charge of genetic analyses in patient 1 and patient 2, respectively. There were no other matches with other phenotypes.

18. HEK293 studies were undertaken as duplicates of n=2. Showing data as mean +/- SEM is inappropriate. Better just show points as a scatter plot (see below).

It was indeed inappropriate to present results of Figure 2D with duplicates of n=2 as mean +/- SEM. We have now repeated this experiment a third time and we could confirm that the two variants identified in patients, p.Thr333Pro and p.Gly430Arg, led to an absence of enzyme activity as compared to the WT protein and to the three other isoforms carrying variants from the general population. We have updated Figure 2D accordingly.

19. "dynamite plunger" plots when numbers are small can hide important data heterogeneity and are now widely disfavoured. Please could all data points be shown superimposed as a scatter plot on these graphs.

As all experiments are now conducted with at least three independent assays and each of them displaying strong homogeneity, we have not added scatter plots to the initial bar graphs.

20. Language: generally good but with small grammatical lapses (e.g. in abstract). Also, even for this clinically qualified reader some of the technical jargon is obscure and not widely used in anglophone medicine. An "ogival palate" would usually be a "high-arched palate" and "spaniomenorrhoea" is better called "oligomenorrhoea". There are other examples of whether more routine medical jargon could be made more accessible for non-clinical readers, too.

We thank Reviewer 3 for his corrections. We have now updated the medical terms as suggested, tried to clarify the manuscript for non-clinical readers, and corrected at least one grammatical mistake in the abstract.